# Understanding and Reducing the Class-Dependent Effects of Data Augmentation with A Two-Player Game Approach

**Yunpeng Jiang**                                                         *jyp9961@sjtu.edu.cn*
*Shanghai Jiao Tong University*

**Yutong Ban**[*]                                                         *yban@sjtu.edu.cn*
*Shanghai Jiao Tong University*

**Paul Weng**[*]                                          *paul.weng@dukekunshan.edu.cn*
*Duke Kunshan University*

**Reviewed on OpenReview:** *https://openreview.net/forum?id=zNsfgCns7x*

## Abstract

Data augmentation is widely applied and has shown its benefits in different machine learning tasks. However, as recently observed, it may have an unfair effect in multi-class classification. While data augmentation generally improves the overall performance (and therefore is beneficial for many classes), it can actually be detrimental for other classes, which can be problematic in some application domains. In this paper, to counteract this phenomenon, we propose CLAM, a CLAss-dependent Multiplicative-weights method. To derive it, we first formulate the training of a classifier as a non-linear optimization problem that aims at simultaneously maximizing the individual class performances and balancing them. By rewriting this optimization problem as an adversarial two-player game, we propose a novel multiplicative weights algorithm, for which we prove the convergence. Interestingly, our formulation also reveals that the class-dependent effects of data augmentation is not due to data augmentation only, but is in fact a general phenomenon. Our empirical results over six datasets demonstrate that the performance of learned classifiers is indeed more fairly distributed over classes, with only limited impact on the average accuracy.

## 1 Introduction

Data augmentation is a popular technique used in the field of computer vision to improve both training efficiency and model performance[1] (Kumar et al., 2023; Ma et al., 2022). It involves creating new training samples by applying semantically-preserving transformations (Shorten & Khoshgoftaar, 2019), such as random shift (Kostrikov et al., 2020; Yarats et al., 2021), random convolution (Hansen et al., 2021) and random resized cropping, to the original data. The aim is to increase the diversity of the training dataset and prevent overfitting, leading to better generalization to the testing dataset. However, recent research (Balestriero et al., 2022) has highlighted a potential issue with data augmentation: although it is beneficial in improving the overall performance, data augmentation can lead to disparities in performance across different classes. This means that while there are significant improvements on the performance of some classes, others may experience a decline. If certain classes are consistently under-performing, it can lead to biased outcomes and a lack of equitable treatment for all classes (Lin et al., 2017), which is important, for instance, in autonomous driving where good label recognition in all classes (e.g., road signs) is crucial to achieve road safety. As a result, there is growing interest in developing methods to achieve equitable performance among all classes within classification tasks, notably to ensure that the benefits of data augmentation are more

---

[*]Corresponding authors.
[1]For concreteness, we focus on accuracy. Our approach could also be applied with other performance measures.

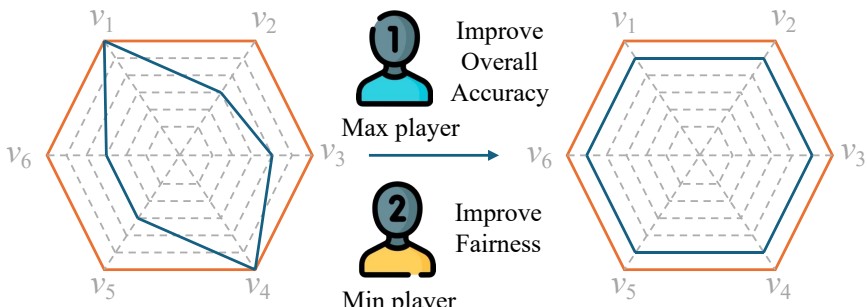

Figure 1: CLAM formulates the classification problem as an adversarial two-player game, where $v_i$ represents the accuracy of the $i^{th}$ class in the classification problem. The min player determines the weights for classes, while the max player aims to maximize the weighted accuracy of the model.

evenly distributed across all classes, rather than being concentrated in a few. This involves finding ways to mitigate the potential negative impact of data augmentation on certain classes while maintaining the positive effects on others.

In this paper, our goal is to address the challenge of reducing the class-dependent effects of data augmentation in classification tasks without much compromise to the overall model performance, aiming to limit the degradation on certain classes caused by data augmentation while maintaining a balance between overall accuracy and class equity. We relate this problem to fair optimization (Ogryczak et al., 2014), which seeks to obtain good solutions with a balanced profile in the context of multi-objective optimization. It has notably been applied in resource allocation problems to ensure a fair treatment of the multiple involved parties. In our work, we apply fair optimization to the realm of classification tasks with data augmentation. As illustrated in Figure 1, our fair optimization problem is formulated as an adversarial zero-sum two-player game. The max player aims at maximizing the overall accuracy of the model while the min player focuses on ensuring fairness over all classes. Instead of the typical max-min fairness formulation, we propose a novel refinement of it to ensure that the accuracies of all classes are optimized, not only that of the worst performing class. In this strategic interaction, the min player determines the weight for each class, assigning larger weights to worse-performing classes, while the max player uses these weights and optimizes the model so that the weighted overall accuracy is maximized. Unlike approaches based on maxmin fairness that concentrate solely on improving the accuracy of the worst-performing class, our strategy ensures that the accuracies of all classes are enhanced, leading to a more balanced distribution of class accuracies. As a side note, our proposed approach actually also extends the popular fair optimization formulation based on the Generalized Gini social welfare function (GGF) (Weymark, 1981), which achieves equity by assigning larger weights to lower-ranked objectives. An important difficulty when using GGF in practice is deciding how to set its fixed weights, which our proposition avoids.

To tackle this two-player game, we propose employing an adaptation of the multiplicative weights algorithm (Freund & Schapire, 1999). Our method dynamically adjusts the weights assigned to each class to ensure that more focus is given to classes with weaker performance. By adopting this multiplicative weights method, we aim to achieve a state of equilibrium where the model's overall accuracy is optimized while simultaneously ensuring fair treatment among all classes. Traditional classification frameworks commonly prioritize overall accuracy while neglecting individual class disparities. By contrast, our innovative method introduces a mechanism that encourages the min player to consider the max player's objective, leading to a trade-off between overall accuracy and class equity.

**Contributions** Our main contributions are three-fold: (i) We propose to reformulate classification using a novel variant of fair optimization to reduce disparities among different classes, which can be viewed as an adversarial zero-sum two-player game. This formalization highlights that the class-dependent effects of data augmentation is actually a more general phenomenon and not only due to data augmentation, which we have also experimentally confirmed. (ii) We propose a multiplicative weights optimization method to solve this

two-player game and further analyze the upper bound of the average loss, which proves the convergence of our method and gives an upper bound on the convergence rate. (iii) Experimental results show that our method mitigates the class-dependent effects of data augmentation, reduces the increase of these effects with stronger data augmentation, enhances the worst class accuracies, and meanwhile maintains a minimal impact on overall average accuracy.

## 2 Related Work

In this section, we introduce related works in fair optimization, fair machine learning, and class equity. Fair optimization integrates fairness-related objectives into the optimization process. In contrast, fair machine learning focuses on developing algorithms that do not discriminate against any particular group based on sensitive attributes. Most relevant to our work, class equity addresses the fairness of a classification model's predictions across all classes.

**Fair Optimization**   In the context of fair optimization (Ogryczak et al., 2014), the Generalized Gini social welfare function (GGF) (Weymark, 1981), which includes the standard max-min fairness (Rawls, 1971), plays a pivotal role in multi-objective settings. The application of GGF for assigning weights is to ensure that the learned model is more equitable and treats all objectives fairly. For instance, in more recent applications in machine learning, Siddique et al. (2020) utilize GGF as a means to guarantee fairness over users in deep reinforcement learning. With the help of GGF, it becomes possible to reduce bias towards any single objective, thus fostering a fairer learning process. More generally, fair social welfare functions have been advocated (Heidari et al., 2018; Speicher et al., 2018; Weng, 2019) as a theoretically-founded approach to tackle fairness in machine learning.

**Fair Machine Learning**   In fair machine learning, previous works consider different definitions of fairness, such as individual fairness and group fairness. Individual fairness focuses on the uniformity of predictions for subjects who possess comparable features relevant to the decision-making process. In group fairness (Krishnaswamy et al., 2021; Jung et al., 2021; Alghamdi et al., 2022; Park et al., 2022; Anchlia & Choi, 2023), the focus is on ensuring that the prediction outcomes for different groups are balanced or equal. In binary classification problems, an algorithm learns a model to predict a target variable and there is a sensitive variable that defines the groups within which to measure the fairness. Group fairness considers three main criteria, independence, separation and sufficiency (Caton & Haas, 2020).

In fair machine learning field, some existing works leverage adversarial two-player game strategy. Xu et al. (2019) have trained generative adversarial networks (GAN) designed to concurrently achieve equitable data generation and classification. This is accomplished by co-training the generative model and the classifier through joint adversarial games with discriminators. Martinez et al. (2020) concentrate on group fairness concerning disparities in predictive risk. They conceptualize fairness as a minimax problem, aiming to discover the classifier that offers the smallest maximum group risk among all efficient models. To address this, they propose the Approximate Projection onto Star Sets (APStar) algorithm, which iteratively refines the minimax risk by updating a linear weighting vector. Closest to our work, Agarwal et al. (2018) reduce fair classification to a series of cost-sensitive classification challenges. They reformulate the issue as an adversarial two-player game and resolve it by optimizing the Lagrangian. Even though the above methods use the max-min two player game strategy, our problem and the formulations, detailed in Section 4, are distinct.

More generally, the objective of our fair classification problem is different to these fair machine learning methods. Our emphasis is on obtaining a classifier whose performance is fair with respect to classes. In our context, the sensitive variable is the same as the target. Consequently, the three main criteria, independence, separation, and sufficiency, no longer apply in our scenario. Moreover, our focus is on reducing the class-dependent effects of data augmentation across a significantly larger number of classes rather than achieving fairness among a constrained number of groups.

**Class Equity**   Specialized loss functions have been introduced to achieve class equity. These approaches are general algorithmic fairness methods for class equity, which can be applied independently of data augmentation. Notably, Lin et al. (2017) introduce the focal loss for the task of dense object detection, which strategically

decreases the training loss from correctly classified instances, thus concentrating the model's learning efforts on the challenging samples. This approach aims to achieve instance fairness, which can also be leveraged in our context for class equity. Szabó et al. (2021) propose tilted cross entropy (TCE) loss, which fosters fairness in semantic segmentation tasks by dynamically assigning weights to various classes based on the training losses. Addressing the problem of model pruning, Meyer & Wong (2022) recognize that while pruning might minimally impact overall performance, it can inadvertently introduce biases, leading to performance degradation for certain sample subsets. Inspired by focal loss, the authors propose a performance weighted loss function aimed at mitigating such unfairness. Bitterwolf et al. (2022) extend evaluation metrics to measure the performance on a group of worst classes, leading to a natural approach that applies the max-min formulation to achieve class equity. In our work, we reformulate classification using fair optimization. Similarly to these methods, our approach can reduce the class-dependent effects of data augmentation while remaining generic and applicable regardless of data augmentation.

Another line of work considers label noise (Scott & Zhang, 2020) as a possible cause of class disparities. Thus, one strategy to mitigate the unfairness of per-class accuracy is through the process of relabelling (Kirichenko et al., 2023). Aiming at reducing the class-dependent effects of data augmentation, like in our work, they demonstrate that by leveraging the more precise multi-label annotations available in ImageNet, the typically negative impact of data augmentation on individual class accuracies is substantially reduced. While their work emphasizes the enhancement of label quality, we directly aim at reducing the class-dependent effects of data augmentation by reformulating the maximization of the average accuracy (see Equation (3)) into a max-min problem (see Equation (4)) encoding fairness over classes. Our approach is complementary to their method and can potentially be combined with theirs.

## 3 Background

In this section, we provide a detailed introduction to the generalized Gini social welfare function and the multiplicative weights method, both of which are closely related to our work. Before that, we first introduce a few notations, which will be useful in the following weighting schemes: $\Delta_n = \{\boldsymbol{w} \in [0,1]^n \mid \forall i, w_i \geq 0; \sum_i w_i = 1\}$ denotes the $(n-1)$-simplex which is the set of all non-negative $n$-dimensional vectors $\boldsymbol{w}$ that sums to one. This simplex is the space to which the weight vectors used in our proposed two-player game belongs. For vectors $\boldsymbol{v}, \boldsymbol{v}' \in \mathbb{R}^n$, the notation $\boldsymbol{v} \leq \boldsymbol{v}'$ means that each element of $\boldsymbol{v}$ is less than or equal to the corresponding element in $\boldsymbol{v}$. Denote the symmetric group of order $n$ (i.e., set of all permutations of $n$ elements) with $\mathbb{S}_n$ and let $\boldsymbol{w_s} = (w_{s(1)}, w_{s(2)}, \ldots, w_{s(n)})$ be a reordering of the weight vector $\boldsymbol{w}$ according to the permutation $s$.

**Generalized Gini Social Welfare Function (GGF)** GGF (Weymark, 1981) extends the concept of Gini coefficient, often used to measure income equality, to a measure of social welfare that encodes fairness:

$$GGF_{\boldsymbol{w}}(\boldsymbol{u}) = \sum_j w_j u_{s(j)}, \tag{1}$$

where $\boldsymbol{w} \in \Delta_n$ such that $w_1 > w_2 > ... > w_n$, $\boldsymbol{u} \in \mathbb{R}^n$ is a utility vector, and $s \in \mathbb{S}_n$ is a permutation such that $u_{s(1)} \leq u_{s(2)} \leq ... \leq u_{s(n)}$. As can be seen from its definition, GGF encodes fairness by assigning higher weights to worse-off components. With different settings of $\boldsymbol{w}$, a higher GGF value indicates either higher overall utility or more equitable distribution. Interestingly, by setting $\boldsymbol{w}$ close to $(1, 0, \ldots, 0)$, GGF can encode the classic max-min fairness. One difficulty with exploiting GGF in applications is that it is not always clear how to choose this weight vector $\boldsymbol{w}$, which actually controls a trade-off between efficiency and equity.

**Multiplicative Weights Method** This method (Freund & Schapire, 1999), also known as *hedge*, is a learning algorithm, which can be applied by a player in adversarial two-player games. It updates the probabilities of the player's strategy based on the outcomes of previous timesteps. Without loss of generality, we assume that the player is the min player. The player initializes a mixed strategy, i.e., probability distribution $\boldsymbol{w} \in \Delta_n$ over all her strategies (e.g., with a uniform distribution). Then at each timestep $t$, the player chooses a strategy $i$ sampled according to her current mixed strategy $\boldsymbol{w}^t$ and incurs loss $v_i$ according

to the other player's choice and the payoff matrix. The player then updates her mixed strategy using the multiplicative weights method. The probability of strategy $i$ is updated as:

$$w_i^{t+1} \leftarrow \frac{w_i^t \cdot e^{-\tau \cdot v_i}}{Z_t} \quad \text{with} \quad Z_t = \sum_{i=1}^n w_i^t \cdot e^{-\tau \cdot v_i}, \tag{2}$$

where $\tau > 0$ is a hyperparameter. This process is repeated for multiple rounds until convergence. The hyperparameter $\tau$ controls the update rate of the probabilities. Freund & Schapire (1999) have proved that with an appropriate choice of $\tau$, the mixed strategy is guaranteed to asymptotically suffer an average loss close to the minimum loss achievable by any fixed strategy.

## 4 Methodology

In this section, we introduce our proposed approach, CLAss-dependent Multiplicative-weights ($CLAM$), where the solution should satisfy two generally contradictory objectives. Firstly, the average accuracy over all classes should be maximized. Secondly, the classifier should not favor any particular class over others. For this purpose, we formulate the classification problem as an adversarial two-player game, which motivates our adaptation of the multiplicative weights method. We then provide a theoretical proof of its convergence to an optimal fair classifier. Finally, we conclude this section with the practical implementation of our adapted multiplicative weights method.

### 4.1 Classification as An Adversarial Two-Player Game

To motivate our formulation of the classification problem (see Equation (5)), we first recall the standard classification problem. Traditionally, an $n$-class classification problem is solved by simply maximizing the average performance over all classes:

$$\max_\theta \frac{1}{n} \sum_{i=1}^n v_i(\theta), \tag{3}$$

where $\theta$ is the parameter of the classification model (e.g., a neural network) and $v_i(\theta)$ is the accuracy achieved by the model for the $i^{th}$ class. As usual in deep learning, this optimization problem is solved approximately by training the model (e.g., minimizing an empirical risk expressed with a convex loss such as the cross-entropy loss). Training is usually performed with the assistance of data augmentation, which can significantly improve the sample efficiency.

However, this standard approach often results in a skewed performance distribution over classes, favoring some over others. To counteract this imbalance, a naive approach would be to resort to the standard minmax fairness[2], which can be interpreted as a zero-sum two-player game. This alternative perspective shifts the focus from the average performance to the performance of the worse class[3]:

$$\max_\theta \min_{\boldsymbol{w} \in \Delta_n} \sum_i w_i v_i(\theta). \tag{4}$$

This formulation aims to enhance the worst-class performance by assigning them greater weights. This two-player game can be solved using the multiplicative weights algorithm: At the end of training epoch $t$, the min player chooses $\boldsymbol{w}$, a probability distribution over classes. With the knowledge of $\boldsymbol{w}$, the max player computes her best response: she optimizes the weighted objective function $\sum_i w_i v_i(\theta)$ by updating $\theta$. After the model parameter update at epoch $t + 1$, the loss of the min player is computed by the class accuracy weighted by $\boldsymbol{w}$. Due to the nature of the zero-sum game, the loss of the max player is the negative of the loss incurred by the min player.

Nonetheless, this max-min approach is insufficient. Indeed, the min player would assign the weights $\boldsymbol{w}$ such that the worst class receives a weight of 1 while all the others receive 0. By focusing solely on the worst class,

---

[2]Here, this leads to a max-min formulation since we want to maximize performance.
[3]This is equivalent to $\max_\theta \min_i v_i(\theta)$.

the performance of the other classes is neglected when training the classifier (i.e., max player), leading to sub-optimal outcomes of the classification model as a whole. As a result, another class would achieve the worst performance and at the next iteration, the weight assigned by the min player would shift to it. This oscillating phenomenon is observed in our experiments: the accuracy fluctuates and the training does not converge.

To avoid this extreme case, we instead use a constrained max-min formulation:

$$\max_{\theta} \min_{\boldsymbol{w} \in \Delta_n^c} \sum_i w_i v_i(\theta), \tag{5}$$

where $\Delta_n^c$ is a compact convex set, which is a strict subset of $\Delta_n$ such that $u_{\min} = \min_i \min_{\boldsymbol{w} \in \Delta_n^c} w_i > 0$ (excluding the extreme points of $\Delta_n$). We assume that any mixed strategy of the max player can be represented by $\theta$. Under this assumption, the max-min problem is equivalent to the min-max problem by the Minimax theorem. Depending on applications, different $\Delta_n^c$ could possibly be defined. In contrast to choosing fixed weights as in GGF, we expect that its definition may be easier since one may define $\Delta_n^c$ such that whole ranges of GGF weights are allowed.

Set $\Delta_n^c$ needs to satisfy some properties such that $f(v) = \min_{\boldsymbol{w} \in \Delta_n^c} \sum_i w_i v_i$ models class equity, as stated by the following proposition whose proof can be found in Appendix A:

**Proposition 4.1.** *If $\Delta_n^c$ satisfies the following property: $\forall \boldsymbol{w} \in \Delta_n^c, \forall s \in \mathbb{S}_n, \boldsymbol{w_s} \in \Delta_n^c$, then $f$ is*

- *monotonic: $\forall \boldsymbol{v}, \boldsymbol{v'}, \boldsymbol{v} \leq \boldsymbol{v'} \Rightarrow f(\boldsymbol{v}) \leq f(\boldsymbol{v'})$,*

- *Schur-concave: $\forall \boldsymbol{v}, v_i > v_j, \forall \epsilon > 0, \epsilon < v_i - v_j, f(\boldsymbol{v} + \epsilon \boldsymbol{e_j} - \epsilon \boldsymbol{e_i}) \geq f(\boldsymbol{v})$,*

- *symmetric: $f(\boldsymbol{v}) = f(\boldsymbol{v_s})$,*

*where $\boldsymbol{e_j}$ represents a vector that is 1 at the $j^{th}$ element and 0 otherwise.*

The mononicity property ensures that all the components should be maximized. Schur-concavity states that if we take any two components $v_i$ and $v_j$ of the vector $\boldsymbol{v}$ such that $v_i > v_j$, and we increase $v_j$ by a small amount while decreasing $v_i$ by the same amount $\epsilon$, the value of $f$ will not decrease, which is consistent with the idea that redistributing resources from a more advantaged component to a less advantaged component should not result in a worse outcome. Symmetry states that the value of the function $f$ remains unchanged regardless of how the components of the vector $\boldsymbol{v}$ are permuted, which ensures that the final outcome does not favor any particular component of the vector $\boldsymbol{v}$. In other words, the function $f$ treats all components of $\boldsymbol{v}$ equally, which is a fundamental aspect of fairness.

The refined formulation in Equation (5) ensures that the maximization of the worst-class performance occurs with a controlled set of weights. Then the efforts to enhance the accuracies of the worst classes may not come at the cost of substantial reductions in the performance of other classes. This offers a more balanced approach to improving the worst-class performance while minimizing the impact on the other classes. This constrained max-min formulation defines our fair optimization problem. Interestingly, when contrasting the traditional formulation (Equation (3)) with our proposed one (Equation (5)), it becomes evident that this issue of class disparities is not specifically related to data augmentation but rather a general problem. While data augmentation can exacerbate this effect, the underlying issue still exists even without it. Also, as another consequence of our formulation, this class-dependent disparity in performance would occur regardless of the network architecture used.

Before explaining how this refined problem can be solved by adapting the multiplicative weights method, we further discuss how it is related to fair optimization as modeled by the GGF, which may also help better understand our novel formulation. This refined problem shares a close relationship with the GGF (Equation (1)). By maximizing GGF during training, the model is not only optimized for higher overall accuracy but also for fairness among classes, leading to an increase of the accuracies of the worst classes. Interestingly, GGF can be rewritten as follows:

$$GGF_{\boldsymbol{w}}(\boldsymbol{v}(\theta)) = \min_{s \in \mathbb{S}_n} \sum_i w_{s(i)} v_i(\theta), \tag{6}$$

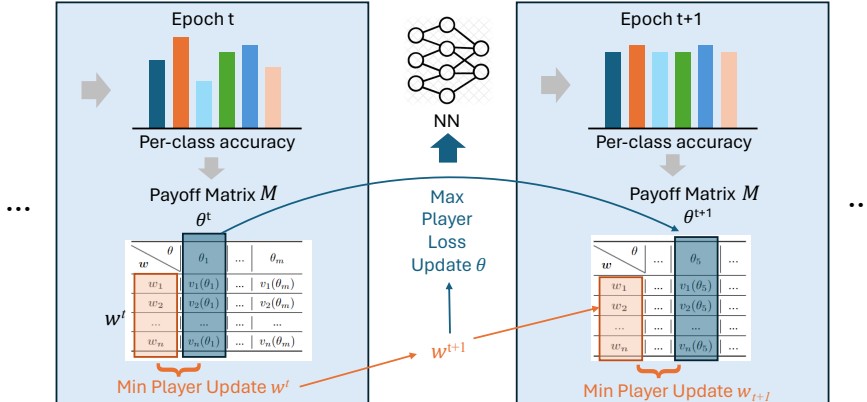

Figure 2: Our method reduces the class-dependent effects of data augmentation iteratively over epochs. At epoch $t$, the max player updates the model parameters $\theta^t$ by a loss function that incorporates weights $\boldsymbol{w_t}$. Subsequently, the min player adjusts the $\boldsymbol{w_t}$ through a multiplicative weights algorithm based on the accuracy of each class, $v_i(\theta^t)$, to obtain the weights for the next epoch $\boldsymbol{w_{t+1}}$. Consequently, the max player loss of epoch $t+1$ is calculated with the updated weights $\boldsymbol{w_{t+1}}$. The above process repeats for each epoch. For the payoff matrix $M$, each row corresponds to a class $i$, each column represents a set of model parameters $\theta$, and each entry $v_i(\theta)$ indicates the accuracy achieved by the model with parameters $\theta$ when evaluated on class $i$.

where $w_{s(i)}$'s are reordered based on $s$. The min over $s$ ensures that at the end, higher $w_{s(i)}$'s are assigned to lower $v_i(\theta)$'s. Using this rewriting, optimizing GGF corresponds to the following max-min problem:

$$\max_{\theta} \min_{s \in \mathbb{S}_n} \sum_i w_{s(i)} v_i(\theta) \tag{7}$$

This problem formulation indicates that when we train the model to optimize the GGF-based objective function, the min player chooses a permutation $s$, while the max player tune the weights $\theta$ to maximize the weighted sum defined by the min player. Therefore, our fair optimization problem defined in Equation (5) includes the GGF-based formulation by setting the constrained simplex to only contain weights that corresponds to reorderings of GGF weights $\boldsymbol{w}$. However, our generic fair optimization problem is more flexible since it does not require to define precisely weight vector $\boldsymbol{w}$, which is hard to choose in practice.

## 4.2 Fair Optimization via Multiplicative Weights Algorithm

As defined above, the min player has a mixed strategy $\boldsymbol{w}$ over classes while the max player maximizes the sum of weighted accuracies by optimizing $\theta$. This adversarial two-player game is well studied. The min player can minimize its loss by applying the multiplicative weights method to attach significance to the worst classes. The max player can directly apply the same weights of all classes $\boldsymbol{w^t}$ as the min player, which ensures that the accuracies of the classes $\boldsymbol{v^t}$ emphasized by the min player are improved most in the next training epoch $t+1$. In Equation (5), we introduce a constrained simplex $\Delta_n^c$. This modification introduces some consequences on the theoretical guarantee, which will be discussed in Section 4.3. The overview of our method is shown in Figure 2. According to this modification, our min-player update rule can be transformed into the following:

$$\boldsymbol{u^t} \leftarrow \boldsymbol{w^t} \times e^{-\tau \cdot \boldsymbol{v^t}}$$
$$\boldsymbol{w^{t+1}} \leftarrow \pi\left(\frac{\boldsymbol{u^t}}{\sum_i u_i^t}\right), \tag{8}$$

where $\tau > 0$ is a hyperparameter and $\pi$ to denote the Euclidean projection on $\Delta_n^c$.

The pseudocode provided in Algorithm 1 outlines the adapted multiplicative weights algorithm. Compared to the standard multiplicative weights algorithm, the key difference lies in the update of the weight vector $\boldsymbol{w^t}$ in line 7 of Algorithm 1. After training the classifier, the accuracies for all classes $\boldsymbol{v^t}$ are used to update $\boldsymbol{w^t}$. It is important to note that $\boldsymbol{w^t}$ is confined to a restricted simplex $\Delta_n^c$ rather than the entire simplex $\Delta_n$.

---

**Algorithm 1** Adapted Multiplicative Weights Algorithm

---

**Hyperparameters**: total number of training epochs $T$, the restricted simplex $\Delta_n^c$.

1: Initialize model parameters $\theta$, a weight vector $\boldsymbol{w^0}$.
2: **for** $t = 0 \ldots T$ **do**
3:     // update $\theta$ (max player)
4:     Update $\theta$ such that $\theta = \arg\max_\theta \sum_i w_i^t v_i(\theta)$.
5:     // update $\boldsymbol{w^t}$ (min player)
6:     Compute the accuracies of all classes $\boldsymbol{v^t}$ at iteration $t$.
7:     Update $\boldsymbol{w^t}$ according to Equation (8).
8: **end for**

---

### 4.3 Theoretical Proof of the Convergence

In this section, we prove that the average loss suffered by the min player in Algorithm 1 is approximately equal to the minimum loss achievable by any fixed strategy, which shows the convergence of Algorithm 1.

**Theorem 1.** *The average loss of the min player's strategy compared to the optimal fixed strategy has an upper bound if we set $\tau$ as $\ln(1 + \frac{1}{1+\max_t \alpha_t}\sqrt{\frac{\ln n}{T}})$.*

$$\frac{1}{T}\sum_{t=1}^{t=T} V_{\boldsymbol{w^t},\boldsymbol{v^t}} \leq \frac{1}{T}\min_{\tilde{\boldsymbol{w}} \in \Delta_n^c}\sum_{t=1}^{t=T} V_{\tilde{\boldsymbol{w}},\boldsymbol{v^t}} + \frac{\ln n}{T} + (1 + \max_t \alpha_t) \cdot \sqrt{\frac{\ln n}{T}}, \tag{9}$$

*where $T$ is the number of total epochs, $\tilde{\boldsymbol{w}}$ can be any arbitrary weight vector in $\Delta_n^c$, $\boldsymbol{w^t}$ is the weight vector chosen by our method at epoch $t$, $\boldsymbol{v^t}$ is the accuracy vector for all classes at epoch $t$, $V_{\boldsymbol{w},\boldsymbol{v}} = \sum_{i=1}^{n} w_i v_i$ is the weighted accuracy of choosing $\boldsymbol{w}$ when the accuracy vector is $\boldsymbol{v}$ and can also be regarded as the loss for the min player, $\alpha_t = \frac{\sum_{i,\epsilon_i^t>0}\tilde{w}_i v_i^t}{\sum_i \tilde{w}_i v_i^t} = \frac{\sum_{i,\epsilon_i^t>0}\tilde{w}_i v_i^t}{V_{\tilde{\boldsymbol{w}},\boldsymbol{v}}}$, $\boldsymbol{x^{t+1}} = \frac{\boldsymbol{u^t}}{\sum_i u_i^t}$, and $\epsilon_i^t = \frac{\pi(\boldsymbol{x^{t+1}})_i}{x_i^{t+1}} - 1$.*

The detailed proof of Theorem 1 can be found in Appendix B. Here we give an outline of the proof. We introduce an intermediate variable $\tilde{\boldsymbol{w}}$ which can be any arbitrary weight vector and compare the difference in KL divergence between consecutive timesteps: $KL(\tilde{\boldsymbol{w}}||\boldsymbol{w^{t+1}}) - KL(\tilde{\boldsymbol{w}}||\boldsymbol{w^t})$. This difference is expressed in terms of the original variables $\boldsymbol{w^t}$ and $\boldsymbol{v^t}$, allowing us to find an upper bound involving $\tau$, $\alpha_t$, $V_{\tilde{\boldsymbol{w}},\boldsymbol{v^t}}$ and $V_{\boldsymbol{w^t},\boldsymbol{v^t}}$. Compared to the proof given by Freund & Schapire (1999), at this step our proof involves another difference term induced by the restricted simplex $\Delta_n^c$, which can also be represented by $\tau$ and $V_{\boldsymbol{w^t},\boldsymbol{v^t}}$. By summing this bound from $t = 1$ to $t = T$, we obtain an overall bound for $KL(\tilde{\boldsymbol{w}}||\boldsymbol{w^{T+1}}) - KL(\tilde{\boldsymbol{w}}||\boldsymbol{w^1})$. Finally, using the property that $\boldsymbol{w^1}$ is a uniform distribution by default and setting a specific value for $\tau$, we derive an upper bound on the average loss of our strategy. This bound gets smaller as $T$ gets larger, showing that our strategy approaches the optimal performance achievable by any fixed strategy in the constrained simplex.

Algorithm 1 suggests that the min player should play the average (over time) of the $\boldsymbol{w}^t$'s as its stationary strategy, which would then yield a near-optimal fair classifier. However, this would require costly computation to obtain it. Instead, we observe that if weight vector $\boldsymbol{w}^t$ converges, then using the last iteration is sufficient. We formalize this point in Lemma 1 in Appendix C.

### 4.4 Practical Implementation

In practice, the restricted simplex $\Delta_n^c$ in Algorithm 1 is selected such that the ratio between the maximum and minimum class weights is lower than a given value. This is to ensure that the max-min approach does not converge to an extreme point, where all probability weight is assigned to one worst class and zero probability to all the others. Proposition 4.1 requires that when the order of weights in the weight vector $\boldsymbol{w} \in \Delta_n^c$ changes, we can still find the modified weight vector $\boldsymbol{w}_s$ in $\Delta_n^c$. The $\Delta_n^c$ we choose in practice does not restrict the order of weights in the weight vector so that it always satisfies Proposition 4.1 in diverse scenarios.

---

**Algorithm 2** CLass-dependent Multiplicative-weights Algorithm (CLAM)

---

**Hyperparameters**: total number of training epochs $T$, total number of iterations in every epoch $K$, multiplicative weights update parameter $\tau$, the lower bound of the weights $u_{\min}$.

 1: Initialize model parameters $\theta$, a weight vector $\boldsymbol{w^0}$.
 2: **for** $t = 0 \ldots T$ **do**
 3:     // update $\theta$
 4:     **for** $k = 0 \ldots K$ **do**
 5:         Sample a minibatch of samples from training set and compute loss $L_i^t$ for class $i$.
 6:         Update the model $\theta$ with the gradients of the loss $L_\theta = \sum_{i=1}^n w_i^t \cdot L_i^t$.
 7:     **end for**
 8:     // update $\boldsymbol{w^t}$
 9:     Compute the training accuracy of all classes $\boldsymbol{v^t}$ at epoch $t$.
10:     Update $\boldsymbol{w^t}$ using $\tau$, $u_{\min}$ and $\boldsymbol{v^t}$ via Equation (8).
11: **end for**

---

A practical implementation of the adapted multiplicative weights algorithm is essential due to the realization of optimization through gradient descent. In the theoretical formulation, the algorithm involves updating $\theta$ to maximize the weighted training accuracies. However, this theoretical approach requires finding an exact solution, which can be computationally infeasible for large-scale problems and can actually also be undesirable due to overfitting issues. To address this challenge, we approximately maximize the weighted training accuracy by updating $\theta$ using gradient descent. The pseudocode in Algorithm 2 describes the practical implementation of Algorithm 1. Note that in contrast to Algorithm 1 where the max player $\theta$ aims to maximize the weighted training accuracy, in Algorithm 2, a loss function is used as a surrogate due to its continuous and differentiable nature, enabling gradient-based optimization for $\theta$.

Theoretically, to ensure unbiased estimation, we should update the weight vector based on accuracies evaluated using a held-out validation dataset. However, empirical observations indicate that this approach can lead to decreased test accuracy. We conjecture that this could be due to the reduced size of the training dataset and possibly higher variance of accuracy estimates in the validation dataset. Consequently, in our proposed method, we use the accuracy in the training dataset to update the weight vector.

## 5 Experimental Results

### 5.1 Baselines, Setups, and Evaluation Metrics

To investigate whether our method alleviate the class-dependent effects of data augmentation, we perform a set of experiments comparing our method with baselines across six different classification tasks.

**Baselines**   Except for normal cross entropy loss (Normal), we further select four baseline methods, which are specifically designed for training a fair classifier. These include focal loss (Focal) (Lin et al., 2017), tilted cross entropy loss (TCE) (Szabó et al., 2021), performance weighted loss (PW) (Meyer & Wong, 2022) and GGF-enhanced cross entropy loss (GGF) (Weymark, 1981; Siddique et al., 2020). The detailed descriptions and hyperparameters are listed in Appendices E and G.

**Experimental Setup**   To evaluate the performance of our method, we run experiments on six classification tasks: CIFAR-10, CIFAR-100 (Krizhevsky & Hinton, 2009), Fashion-Mnist (Xiao et al., 2017), iNaturalist2018 (Horn et al., 2018), Mini-ImageNet and ImageNet (Russakovsky et al., 2015). Regarding the network architecture, since the issue of class-dependent effects is brought up by Balestriero et al. (2022) where they employ ResNet-50 models, we follow their experimental setup and train a ResNet-50 model from scratch as well. As discussed in Section 4.1, our mathematical formulation reveals that this issue and our proposed method are independent of the neural architecture of the classifier and one may expect similar results with other architectures (e.g., ViT (Dosovitskiy et al., 2021)). Either random resized cropping or color jittering is applied during training. For each method and each task, we run experiments with 8 data augmentation parameters,

Table 1: We present the fairness metrics of different methods on five datasets in the following two tables. In the second column of each table, "std", "COV", and "range" respectively denote the standard deviation, coefficient of variation, and range of class accuracies on test datasets. For each method used on each dataset, the reported result is averaged over 8 crop lower bounds. The std and range are listed in % while COV is listed in values. **Lower is better.**

| | CIFAR-10 | | | CIFAR-100 | | | Fashion-Mnist | | | Mini-ImageNet | | | ImageNet | | |
|---|---|---|---|---|---|---|---|---|---|---|---|---|---|---|---|
| Method | std | COV | range | std | COV | range | std | COV | range | std | COV | range | std | COV | range |
| Normal | 4.55 | 0.049 | 12.6 | 11.8 | 0.157 | 39.1 | 5.94 | 0.063 | 18.3 | 11.2 | 0.152 | 37.0 | 17.3 | 0.247 | 58.5 |
| Focal | 4.76 | 0.052 | 14.1 | 11.3 | 0.150 | 37.2 | 6.05 | 0.064 | 18.9 | 11.2 | 0.151 | 36.6 | 17.0 | 0.252 | 57.7 |
| PW | 4.65 | 0.050 | 13.4 | 11.4 | 0.152 | 37.8 | 6.12 | 0.066 | 18.7 | 11.2 | 0.151 | 36.5 | 17.4 | 0.256 | 58.7 |
| TCE | 4.56 | 0.049 | 13.0 | 11.6 | 0.156 | 38.7 | 6.14 | 0.066 | 19.0 | 11.4 | 0.155 | 37.2 | 17.1 | 0.245 | 58.0 |
| GGF | 4.12 | 0.044 | 11.9 | 11.2 | 0.149 | 36.9 | 5.98 | 0.064 | 19.4 | 12.1 | 0.172 | 41.3 | 16.9 | 0.242 | 57.2 |
| **CLAM (Ours)** | **3.86** | **0.042** | **10.7** | **10.8** | **0.143** | **35.9** | **5.44** | **0.058** | **16.7** | **10.6** | **0.143** | **35.1** | **16.7** | **0.241** | **56.8** |

Figure 3: Fairness metrics over training epochs on CIFAR-100 and Mini-Imagenet. "std" and "COV" denote the standard deviation and coefficient of variation for evaluation accuracies. (**Lower is better.**)

crop lower bound for random resized cropping, and brightness, contrast, and saturation adjustment for color jittering.

**Evaluation Metrics**   To investigate whether our method can alleviate the class-dependent effects of data augmentation, we calculate the range, standard deviation, and the coefficient of variation (COV) of class accuracies on test datasets. COV measures relative variability and is calculated as the ratio of the standard deviation to the mean. We also compare the worst class accuracies achieved by our method on test datasets with those of the baseline methods. The mathematical definitions of these metrics can be found in Appendix H.

## 5.2   Results and Analysis

Through these experiments, we aim to provide detailed answers to several key questions. First, we investigate whether our method alleviate the class-dependent effects of data augmentation. Next, we assess if our approach reduces the increase of these effects caused by data augmentation. We also examine the improvement in performance for the worst classes. Additionally, we present the evolution of the weight vector during training. Furthermore, we evaluate whether our method significantly degrades the overall performance. Due to the page limit, we present the comparison of our method and group fairness methods in Appendix D, and the hyperparameter sensitivity study on $\tau$ in Appendix L.

**Does our method alleviate the class-dependent effects of data augmentation?**   To effectively address whether our method alleviates the class-dependent effects of data augmentation, we present the results from five datasets in Table 1. Consistently across all five datasets, our method leads to a remarkable reduction in the standard deviation and range of class accuracies. This reduction indicates a narrower spread of accuracies among different classes, which is a positive indicator for fairness. The coefficient of variation further corroborates the trend by demonstrating a reduced relative variability in class accuracies,

Table 2: Difference in test accuracy range (with DA - without DA) across five datasets, averaged over seven crop lower bounds with data augmentation. Lower values indicate lower increase of class-dependent effects of data augmentation. The difference values are listed in %. **Lower is better.**

|  | CIFAR10 | CIFAR100 | FMnist | MImageNet | ImageNet |
|---|---|---|---|---|---|
| Method | Difference in Test Accuracy Range (mean ± std) | | | | |
| Normal (2007) | 1.0±1.0 | 0.5±0.6 | 0.5±0.9 | 0.6±0.8 | -3.0±0.5 |
| Focal (ICCV2017) | 1.7±0.5 | -1.1±0.5 | 2.3±0.7 | 1.5±0.7 | -3.5±0.6 |
| PW (NeurIPS2022) | -0.5±0.6 | 2.9±0.8 | 1.1±1.8 | 0.1±0.6 | 0.0±0.8 |
| TCE (CVPR2021) | -1.2±0.5 | -1.8±0.5 | 0.5±1.6 | **-5.2**±0.2 | -1.5±0.6 |
| GGF (ICML2020) | -1.7±0.6 | 0.3±0.7 | -0.7±1.1 | 0.4±1.7 | 0.0±1.0 |
| **CLAM**(Ours) | **-2.4**±0.3 | **-2.3**±0.7 | **-1.4**±1.4 | -3.5±0.9 | **-4.0**±0.8 |

Table 3: The worst class test accuracies of different methods on five datasets, calculated from the worst 10% classes and averaged over 8 crop lower bounds. The accuracies are listed in %. **Higher is better.**

|  | CIFAR10 | CIFAR100 | FMnist | MImageNet | ImageNet |
|---|---|---|---|---|---|
| Method | Worst Class Test Accuracies (mean ± std) | | | | |
| Normal (2007) | 84.4±1.9 | 53.6±3.0 | 81.0±1.6 | 53.8±4.5 | 35.9±2.6 |
| Focal (ICCV2017) | 82.4±1.1 | 55.1±2.4 | 80.5±1.4 | 52.9±4.2 | 34.8±2.1 |
| PW (NeurIPS2022) | 83.4±1.3 | 54.7±3.0 | 80.6±3.6 | 53.9±3.8 | 34.4±2.1 |
| TCE (CVPR2021) | 84.3±1.4 | 53.1±3.7 | 80.2±3.4 | 52.8±3.2 | 36.2±2.5 |
| GGF (ICML2020) | 85.1±1.9 | 55.5±3.1 | 80.0±2.0 | 47.7±3.3 | 36.6±2.2 |
| **CLAM**(Ours) | **85.8**±1.4 | **56.4**±2.3 | **82.5**±2.6 | **54.8**±3.7 | **36.7**±2.9 |

which implies that the distribution of class accuracies is more uniform. Results of color jittering as a data augmentation technique are included in Appendix M. Additional results on iNaturalist2018 and ImageNet with reassessed labels (Beyer et al., 2020; Kirichenko et al., 2023) are provided in Appendices N and O. These results strongly support the assertion that our method learns a classifier that exhibits a more balanced distribution of accuracies across classes. Plots of fairness metrics along the training process are included in Figure 3. We can observe that our method consistently outperforms others with lower standard deviation and COV along the training process. Due to the page limit, plots of other datasets are included in Appendix J.

**Does our method reduce the increase of class-dependent effects of data augmentation?** In this part, we aim to quantify the effectiveness of our method in reducing the increase of the class-dependent effects associated with data augmentation. We measure the increase of class-dependent effects by calculating the difference in accuracy ranges when data augmentation is applied versus when it is not. A smaller difference value indicates that data augmentation leads to a reduced accuracy range. A larger difference value suggests an increased accuracy range, implying a higher increase of class-dependent effects of data augmentation, which we aim to avoid. Table 2 illustrates that our method consistently yields the lowest difference values. This observation suggests that our method effectively reduces the increase of these effects induced by data augmentation Therefore, our approach ensures that the performance gains from data augmentation are not disproportionately influenced by the characteristics of certain classes, leading to a more balanced and reliable enhancement in accuracy across all classes.

**Does our method improve the worst class performance?** One overarching goal of our method is to ensure that the performances of the least accurate classes receive some enhancement. The accuracies of the worst 10% classes across five diverse datasets are shown in Table 3. We can observe a consistent trend of improvement of the worst-class accuracies with our method across all five datasets, indicating that our

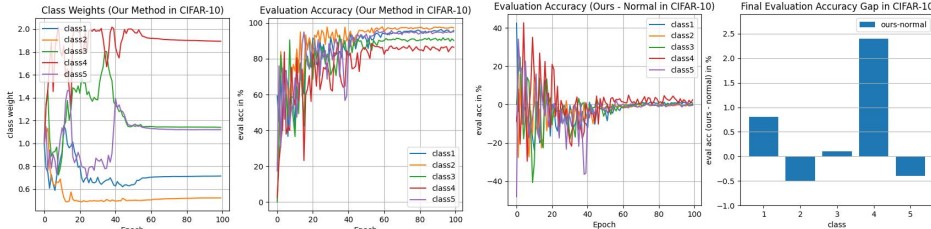

Figure 4: Evolution of class weights in CIFAR-10. From left to right: class weights, evaluation accuracy, evaluation accuracy gap during training, and final evaluation accuracy gap.

Table 4: The average test accuracies of different methods on five datasets. We average the results over 8 crop lower bounds. The accuracies are listed in %. **Higher is better.**

|  | CIFAR10 | CIFAR100 | FMnist | MImageNet | ImageNet |
|---|---|---|---|---|---|
| Method | Average Test Accuracies (mean ± std) | | | | |
| Normal (2007) | 92.8±0.6 | 75.2±2.0 | 94.1±0.5 | 74.4±2.6 | **70.3**±2.6 |
| Focal (ICCV2017) | 91.9±0.5 | 75.5±1.6 | 94.0±0.3 | 74.2±2.8 | 67.7±2.5 |
| PW (NeurIPS2022) | 92.4±0.6 | 75.2±1.8 | 93.6±1.3 | 74.3±3.0 | 68.0±2.4 |
| TCE (CVPR2021) | 92.8±0.8 | 74.7±2.4 | 93.7±0.9 | 73.7±2.2 | 70.2±2.3 |
| GGF (ICML2020) | **93.0**±1.2 | 75.3±1.8 | **94.2**±0.4 | 70.6±2.1 | 70.0±2.2 |
| **CLAM**(Ours) | 92.6±0.9 | **75.7**±1.7 | 94.0±0.4 | **74.5**±2.3 | 69.6±2.4 |

approach is effective in mitigating the disparity in class accuracies, resulting in more balanced classifiers. To visualize the enhancements achieved by our method, we provide plots of worst class accuracies in Appendix I.

**How does the weight vector evolve during training?** We show the evolution of the weight vector during training. In the loss function we normalize the weights such that the sum equals $n$ rather than 1, a deliberate implementation choice that does not affect the order of weights. We present the changes of the weights for certain classes in the CIFAR-10 dataset, as shown in Figure 4. We can observe that the order of the weights alternates during the initial phase of training. Then they begin to converge around the $45^{th}$ epoch, and subsequently the test accuracies converge at approximately the $60^{th}$ epoch. Notably, the order of the converged weights is inversely related to the order of the converged test accuracies, which is desired. Plots of the other tasks are provided in Appendix K.

**Does our method significantly degrade the overall performance?** We display the average test accuracies in Table 4. In three out of the five datasets, our method results in a slight decrease in overall accuracy compared to the baselines. However, this degradation is relatively limited compared to our improvement of worst class accuracies, suggesting that our approach manages to balance the contradictory demands of accuracy and equity. Remarkably, in the remaining two datasets, our method surpasses all baselines, underscoring its potential to enhance both class equity and overall accuracy simultaneously.

## 6 Conclusion

In this paper, we focus on the class-dependent effects of data augmentation in classification. We propose CLAM, a novel approach that reformulates classification as a two-player game and adjusts class weights to achieve a balance between class equity and overall accuracy. CLAM is validated both empirically and theoretically. Importantly, our method is effective with very few hyperparameters. This simplicity makes it easily adaptable to various other machine learning applications. For future work, it could potentially be applied to other scenarios, such as multi-task problems to achieve an equitable distribution of performance across various tasks.

## Acknowledgments

This work has been supported by the program of National Natural Science Foundation of China (No. 62176154) and by Shanghai Magnolia Funding Pujiang Program (No. 23PJ1404400). We would also like to thank Yann Chevaleyre for insightful discussions.

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

## A  Proof of Proposition 4.1

Let $f(v) = \min_{\boldsymbol{w} \in \Delta_n^c} \sum_i w_i v_i$. If $\Delta_n^c$ satisfies the following property: $\forall \boldsymbol{w} \in \Delta_n^c, \forall s \in \mathbb{S}_n, \boldsymbol{w_s} \in \Delta_n^c$, where $\mathbb{S}_n$ is the symmetric group of order $n$, i.e., set of all permutations of $n$ elements and $\boldsymbol{w_s}$ is a reordering of the weight vector $\boldsymbol{w}$ based on $s$, then $f$ is

- monotonic: $\forall \boldsymbol{v}, \boldsymbol{v'}, \boldsymbol{v} \leq \boldsymbol{v'} \Rightarrow f(\boldsymbol{v}) \leq f(\boldsymbol{v'})$,

- Schur-concave: $\forall \boldsymbol{v}, v_i > v_j, \forall \epsilon > 0, \epsilon < v_i - v_j, f(\boldsymbol{v} + \epsilon \boldsymbol{e_j} - \epsilon \boldsymbol{e_i}) \geq f(\boldsymbol{v})$,

- symmetric: $f(\boldsymbol{v}) = f(\boldsymbol{v_s})$,

where $\boldsymbol{e_j}$ represents a vector that is 1 at the $j^{th}$ element and 0 otherwise.

*Proof.* Firstly, it is obvious that $f$ is symmetric. $f(\boldsymbol{v}) = f(\boldsymbol{v_s})$ because $\forall \boldsymbol{w} \in \Delta_n^c, \forall s, \boldsymbol{w_s} \in \Delta_n^c$.

Then, we prove that $f$ is monotonic. Let $\boldsymbol{w'} = \arg\min_{\boldsymbol{w} \in \Delta_n^c} \boldsymbol{w} \cdot \boldsymbol{v'}$.

$$
\begin{aligned}
f(\boldsymbol{v'}) &= \min_{\boldsymbol{w} \in \Delta_n^c} \boldsymbol{w} \cdot \boldsymbol{v'} \\
&= \boldsymbol{w'} \cdot \boldsymbol{v'} \\
&\geq \boldsymbol{w'} \cdot \boldsymbol{v} \\
&\geq \min_{\boldsymbol{w} \in \Delta_n^c} \boldsymbol{w} \cdot \boldsymbol{v} \\
&= f(v)
\end{aligned}
\tag{10}
$$

The first inequality holds because $\boldsymbol{v'} \geq \boldsymbol{v}$. The second inequality holds by definition.

Finally, we prove that $f$ is Schur-concave. Let $\boldsymbol{y} = \arg\min_{\boldsymbol{w} \in \Delta_n^c} \boldsymbol{w} \cdot (\boldsymbol{v} + \epsilon \boldsymbol{e_j} - \epsilon \boldsymbol{e_i})$. There are two scenarios depending on the value of $\epsilon$.

If $0 < \epsilon \leq \frac{1}{2}(v_i - v_j)$, then $v_j + \epsilon \leq v_i - \epsilon$. By definition of $\boldsymbol{y}$, $y_j \geq y_i$.

$$
\begin{aligned}
f(\boldsymbol{v} + \epsilon \boldsymbol{e_j} - \epsilon \boldsymbol{e_i}) &= \min_{\boldsymbol{w} \in \Delta_n^c} \boldsymbol{w} \cdot (\boldsymbol{v} + \epsilon \boldsymbol{e_j} - \epsilon \boldsymbol{e_i}) \\
&= \boldsymbol{y} \cdot (\boldsymbol{v} + \epsilon \boldsymbol{e_j} - \epsilon \boldsymbol{e_i}) \\
&= y_i \cdot (v_i - \epsilon) + y_j \cdot (v_j + \epsilon) + \sum_{k \neq i,j} y_k \cdot v_k \\
&= \epsilon \cdot (y_j - y_i) + \sum_k y_k \cdot v_k \\
&\geq \sum_k y_k \cdot v_k \\
&\geq \min_{\boldsymbol{w} \in \Delta_n^c} \boldsymbol{w} \cdot \boldsymbol{v} \\
&= f(\boldsymbol{v})
\end{aligned}
\tag{11}
$$

If $\frac{1}{2}(v_i - v_j) \leq \epsilon < v_i - v_j$, we introduce an intermediate variable $\epsilon' = v_i - v_j - \epsilon$, satisfying $0 < \epsilon' \leq \frac{1}{2}(v_i - v_j)$. $v_j + \epsilon' = v_i - \epsilon$ and $v_i - \epsilon' = v_j + \epsilon$. Since $f$ is symmetric,

$$
\begin{aligned}
f(\boldsymbol{v} + \epsilon \boldsymbol{e_j} - \epsilon \boldsymbol{e_i}) &= f(\boldsymbol{v} + \epsilon' \boldsymbol{e_j} - \epsilon' \boldsymbol{e_i}) \\
&\geq f(v)
\end{aligned}
\tag{12}
$$

Therefore, $f(\boldsymbol{v} + \epsilon \boldsymbol{e_j} - \epsilon \boldsymbol{e_i}) \geq f(\boldsymbol{v})$. $\qquad \square$

## B  Proof of Theorem 1

The average loss of the min player's strategy compared to the optimal fixed strategy has an upper bound if we set $\tau$ as $\ln(1 + \frac{1}{1+\max\limits_{t}\alpha_t}\sqrt{\frac{\ln n}{T}})$.

$$\frac{1}{T}\sum_{t=1}^{t=T} V_{\boldsymbol{w^t},\boldsymbol{v^t}} \leq \frac{1}{T}\min_{\tilde{\boldsymbol{w}}\in\Delta_n^c}\sum_{t=1}^{t=T} V_{\tilde{\boldsymbol{w}},\boldsymbol{v^t}} + \frac{\ln n}{T}$$
$$+ (1 + \max_{t}\alpha_t)\cdot\sqrt{\frac{\ln n}{T}} \tag{13}$$

where $T$ is the number of total epochs, $\tilde{\boldsymbol{w}}$ can be any arbitrary weight vector, $\boldsymbol{w^t}$ is the weight vector chosen by our method at epoch $t$, $\boldsymbol{v^t}$ is the accuracy vector for all classes at epoch $t$, $V_{\boldsymbol{w},\boldsymbol{v}} = \sum_{i=1}^{n} w_i v_i$ is the weighted accuracy of choosing $w$ when the accuracy vector is $\boldsymbol{v}$ and can also be regarded as the loss for the min player, $\alpha_t = \frac{\sum_{i,\epsilon_i^t>0}\tilde{w}_i v_i^t}{\sum_i \tilde{w}_i v_i^t} = \frac{\sum_{i,\epsilon_i^t>0}\tilde{w}_i v_i^t}{V_{\tilde{\boldsymbol{w}},\boldsymbol{v}}}$, $\boldsymbol{x^{t+1}} = \frac{\boldsymbol{u^t}}{\sum_i u_i^t}$, and $\epsilon_i^t = \frac{\pi(\boldsymbol{x^{t+1}})_i}{x_i^{t+1}} - 1$.

*Proof.* We introduce an intermediate variable $\boldsymbol{x^{t+1}} = \frac{\boldsymbol{u^t}}{\sum_i \boldsymbol{u_i^t}}$. Then our update rule can be rewritten as follows.

$$\boldsymbol{u^t} \leftarrow \boldsymbol{w^t}\times e^{-\tau\cdot\boldsymbol{v^t}},$$
$$\boldsymbol{w^{t+1}} \leftarrow \pi(\boldsymbol{x^{t+1}}). \tag{14}$$

We begin the proof by analyzing $KL(\tilde{\boldsymbol{w}}||\boldsymbol{w^{t+1}}) - KL(\tilde{\boldsymbol{w}}||\boldsymbol{w^t})$, where $\tilde{\boldsymbol{w}}$ is any arbitrary weight vector.

$$KL(\tilde{\boldsymbol{w}}||\boldsymbol{w^{t+1}}) - KL(\tilde{\boldsymbol{w}}||\boldsymbol{w^t})$$
$$= [KL(\tilde{\boldsymbol{w}}||\boldsymbol{w^{t+1}}) - KL(\tilde{\boldsymbol{w}}||\boldsymbol{x^{t+1}})]$$
$$+ [KL(\tilde{\boldsymbol{w}}||\boldsymbol{x^{t+1}}) - KL(\tilde{\boldsymbol{w}}||\boldsymbol{w^t})] \tag{15}$$

We firstly prove that for any $\tilde{\boldsymbol{w}}$, $KL(\tilde{\boldsymbol{w}}||\boldsymbol{w^{t+1}}) - KL(\tilde{\boldsymbol{w}}||\boldsymbol{x^{t+1}})$ has an upper bound.

Let

$$\epsilon_i^t = \frac{\pi(\boldsymbol{x^{t+1}})_i}{\boldsymbol{x_i^{t+1}}} - 1$$
$$\alpha_t = \frac{\sum_{i,\epsilon_i^t>0}\tilde{w}_i v_i^t}{\sum_i \tilde{w}_i v_i^t} = \frac{\sum_{i,\epsilon_i^t>0}\tilde{w}_i v_i^t}{V_{\tilde{\boldsymbol{w}},\boldsymbol{v}}} \tag{16}$$

$$KL(\tilde{\boldsymbol{w}}||\boldsymbol{w^{t+1}}) - KL(\tilde{\boldsymbol{w}}||\boldsymbol{x^{t+1}})$$
$$= \sum_{i=1}^{i=n} \tilde{w}_i \ln\frac{\boldsymbol{x_i^{t+1}}}{\boldsymbol{w_i^{t+1}}}$$
$$= \sum_{i=1}^{i=n} \tilde{w}_i \ln\frac{\boldsymbol{x_i^{t+1}}}{\pi(\boldsymbol{x^{t+1}})_i} \tag{17}$$
$$\leq \sum_{i,\epsilon_i^t>0} \tilde{w}_i\tau v_i^t = \alpha_t\cdot\tau V_{\tilde{\boldsymbol{w}},\boldsymbol{v}}$$

The inequality holds because $\ln\frac{\pi(\boldsymbol{x^{t+1}})_i}{\boldsymbol{x_i^{t+1}}} = 0$ for $\epsilon_i^t = 0$ and $\pi(\boldsymbol{x^{t+1}})_i = \frac{\max(x_i^{t+1},x_{\min})}{Z_x} = \frac{x_{\min}}{Z_x} \geq x_{\min}\cdot e^{-\tau\cdot v_i^t} \geq$

$x_i^{t+1}\cdot e^{-\tau\cdot v_i^t}$ for $\epsilon_i^t > 0$, where $Z_x = \sum_{i=1}^{i=n}\max(x_i^{t+1},x_{\min}) = \frac{\sum_{i=1}^{i=n}\max(x_i^{t+1},x_{\min})}{\sum_{i=1}^{i=n}x_i^{t+1}} \leq \frac{\sum_{i,\epsilon_i^t>0}\max(x_i^{t+1},x_{\min})}{\sum_{i,\epsilon_i^t>0}x_i^{t+1}} \leq$

$\frac{\sum_{i,\epsilon_i^t>0}x_{\min}}{\sum_{i,\epsilon_i^t>0}x_{\min}\cdot e^{-\tau\cdot\min(v_i^t)}} \leq e^{\tau\cdot\min(v_i^t)}$.

We then prove that for any $\tilde{\boldsymbol{w}}$, $KL(\tilde{\boldsymbol{w}}||\boldsymbol{x^{t+1}}) - KL(\tilde{\boldsymbol{w}}||\boldsymbol{w^t})$ has an upper bound. For simplicity, define $V_{\boldsymbol{w},\boldsymbol{v}} = \sum_i w_i v_i$ and $Z_t = \sum_i u_i^t$.

$$
\begin{aligned}
& KL(\tilde{\boldsymbol{w}}||\boldsymbol{x^{t+1}}) - KL(\tilde{\boldsymbol{w}}||\boldsymbol{w^t}) \\
&= \sum_{i=1}^{i=n} \tilde{w}_i \ln \frac{w_i^t}{x_i^{t+1}} \\
&= \sum_{i=1}^{i=n} \tilde{w}_i \ln \left( \frac{Z_t}{e^{-\tau \cdot v_i^t}} \right) \\
&= \sum_{i=1}^{i=n} \tilde{w}_i \ln Z_t + \tau \sum_{i=1}^{i=n} \tilde{w}_i v_i^t \\
&= \ln Z_t + \tau V_{\tilde{\boldsymbol{w}},\boldsymbol{v^t}} \\
&= \ln \sum_{i=1}^{i=n} w_i^t \cdot e^{-\tau \cdot v_i^t} + \tau V_{\tilde{\boldsymbol{w}},\boldsymbol{v^t}} \\
&\leq \ln \sum_{i=1}^{i=n} w_i^t \cdot (1 - (1 - e^{-\tau}) v_i^t) + \tau V_{\tilde{\boldsymbol{w}},\boldsymbol{v^t}} \\
&= \ln \left( 1 - (1 - e^{-\tau}) \sum_{i=1}^{i=n} w_i^t \cdot v_i^t \right) + \tau V_{\tilde{\boldsymbol{w}},\boldsymbol{v^t}} \\
&= \ln(1 - (1 - e^{-\tau}) V_{\boldsymbol{w^t},\boldsymbol{v^t}}) + \tau V_{\tilde{\boldsymbol{w}},\boldsymbol{v^t}} \\
&\leq -(1 - e^{-\tau}) V_{\boldsymbol{w^t},\boldsymbol{v^t}} + \tau V_{\tilde{\boldsymbol{w}},\boldsymbol{v^t}}
\end{aligned}
\tag{18}
$$

The first inequality holds because $e^{-\tau v_i^t} <= 1 - (1 - e^{-\tau}) v_i^t$ for $\tau > 0$ and $0 <= v_i^t <= 1$. The second inequality holds because $\ln(1 - (1 - e^{-\tau}) V_{\boldsymbol{w^t},\boldsymbol{v^t}}) \leq -(1 - e^{-\tau}) V_{\boldsymbol{w^t},\boldsymbol{v^t}}$ for $0 <= (1 - e^{-\tau}) V_{\boldsymbol{w^t},\boldsymbol{v^t}} < 1$.

Sum eq. (17) and eq. (18) together, we get the upper bound of $KL(\tilde{\boldsymbol{w}}||\boldsymbol{w^{t+1}}) - KL(\tilde{\boldsymbol{w}}||\boldsymbol{w^t})$.

$$
\begin{aligned}
& KL(\tilde{\boldsymbol{w}}||\boldsymbol{w^{t+1}}) - KL(\tilde{\boldsymbol{w}}||\boldsymbol{w^t}) \\
&\leq -(1 - e^{-\tau}) V_{\boldsymbol{w^t},\boldsymbol{v^t}} + (1 + \alpha_t) \tau V_{\tilde{\boldsymbol{w}},\boldsymbol{v^t}}
\end{aligned}
\tag{19}
$$

Sum eq. (19) from $t = 1$ to $t = T$, we get the upper bound of $KL(\tilde{\boldsymbol{w}}||\boldsymbol{w^{T+1}}) - KL(\tilde{\boldsymbol{w}}||\boldsymbol{w^1})$.

$$
\begin{aligned}
& KL(\tilde{\boldsymbol{w}}||\boldsymbol{w^{T+1}}) - KL(\tilde{\boldsymbol{w}}||\boldsymbol{w^1}) \\
&\leq \sum_{t=1}^{t=T} \left( -(1 - e^{-\tau}) V_{\boldsymbol{w^t},\boldsymbol{v^t}} + (1 + \alpha_t) \tau V_{\tilde{\boldsymbol{w}},\boldsymbol{v^t}} \right)
\end{aligned}
\tag{20}
$$

From eq. (20), we get the upper bound of $\sum_{t=1}^{t=T} V_{\boldsymbol{w^t},\boldsymbol{v^t}}$. Note that since $w_i^1 = \frac{1}{n}$ for all $i$, $KL(\tilde{\boldsymbol{w}}||\boldsymbol{w^{T+1}}) - KL(\tilde{\boldsymbol{w}}||\boldsymbol{w^1}) \leq \ln n$ for any $\tilde{\boldsymbol{w}}$. Here, we set $\tau$ as $\ln(1 + \frac{1}{1+\max\limits_t \alpha_t} \sqrt{\frac{\ln n}{T}})$.

$$
\begin{aligned}
\sum_{t=1}^{t=T} V_{\boldsymbol{w^t},\boldsymbol{v^t}} \leq & \frac{1}{1-e^{-\tau}} \Big( KL(\boldsymbol{\tilde{w}}||\boldsymbol{w^1}) - KL(\boldsymbol{\tilde{w}}||\boldsymbol{w^{T+1}}) \\
& + (1+\alpha_t)\tau \sum_{t=1}^{t=T} V_{\boldsymbol{\tilde{w}},\boldsymbol{v^t}} \Big) \\
\leq & \frac{1}{1-e^{-\tau}} \ln n \\
& + (1+\max_t \alpha_t)\frac{\tau}{1-e^{-\tau}} \sum_{t=1}^{t=T} V_{\boldsymbol{\tilde{w}},\boldsymbol{v^t}} \\
\leq & (1+\frac{1}{e^{\tau}-1})\ln n \\
& + (1+\max_t \alpha_t)\frac{1+e^{-\tau}}{2e^{-\tau}} \sum_{t=1}^{t=T} V_{\boldsymbol{\tilde{w}},\boldsymbol{v^t}} \\
= & (1+\frac{1}{e^{\tau}-1})\ln n \\
& + (1+\max_t \alpha_t)\frac{1+e^{\tau}}{2} \sum_{t=1}^{t=T} V_{\boldsymbol{\tilde{w}},\boldsymbol{v^t}} \\
= & \left(1 + (1+\max_t \alpha_t)\cdot \sqrt{\frac{T}{\ln n}}\right) \ln n \\
& + (1+\max_t \alpha_t) \\
& \cdot \left(1 + \frac{1}{2(1+\max_t \alpha_t)}\sqrt{\frac{\ln n}{T}}\right) \sum_{t=1}^{t=T} V_{\boldsymbol{\tilde{w}},\boldsymbol{v^t}}
\end{aligned}
\tag{21}
$$

Dividing both sides by $T$, we obtain the upper bound of $\frac{1}{T}\sum_{t=1}^{t=T} V_{\boldsymbol{w^t},\boldsymbol{v^t}}$.

$$
\begin{aligned}
\frac{1}{T}\sum_{t=1}^{t=T} V_{\boldsymbol{w^t},\boldsymbol{v^t}} \leq & \frac{\ln n}{T} + (1+\max_t \alpha_t)\cdot \sqrt{\frac{\ln n}{T}} \\
& + \left(\frac{1+\max_t \alpha_t}{T} + \frac{1}{2}\sqrt{\frac{\ln n}{T^3}}\right) \sum_{t=1}^{t=T} V_{\boldsymbol{\tilde{w}},\boldsymbol{v^t}} \\
\leq & \left(\frac{1+\max_t \alpha_t}{T} + \frac{1}{2}\sqrt{\frac{\ln n}{T^3}}\right) \\
& \cdot \min_{\boldsymbol{\tilde{w}}\in\Delta_n^c}\sum_{t=1}^{t=T} V_{\boldsymbol{\tilde{w}},\boldsymbol{v^t}} \\
& + \frac{\ln n}{T} + (1+\max_t \alpha_t)\cdot \sqrt{\frac{\ln n}{T}} \\
\approx & \left(\frac{1}{T} + \frac{1}{2}\sqrt{\frac{\ln n}{T^3}}\right) \min_{\boldsymbol{\tilde{w}}\in\Delta_n^c}\sum_{t=1}^{t=T} V_{\boldsymbol{\tilde{w}},\boldsymbol{v^t}} \\
& + \frac{\ln n}{T} + (1+\max_t \alpha_t)\cdot \sqrt{\frac{\ln n}{T}}
\end{aligned}
\tag{22}
$$

$\frac{1}{T}\sum_{t=1}^{t=T} V_{\boldsymbol{w^t},\boldsymbol{v^t}}$ represents the average performance of our strategy while $\min_{\boldsymbol{\tilde{w}}\in\Delta_n^c}\frac{1}{T}\sum_{t=1}^{t=T} V_{\boldsymbol{\tilde{w}},\boldsymbol{v^t}}$ represents the best performance we can achieve by any fixed $\boldsymbol{\tilde{w}}$ that is in the constrained simplex . The above inequality

guarantees that the average loss of our strategy compared to this optimal $\tilde{\boldsymbol{w}}$ is approximately bounded by $\frac{\ln n}{T} + (1 + \max_t \alpha_t) \cdot \sqrt{\frac{\ln n}{T}}$, which can be minimized when $T$ gets larger. $\hspace{1cm}\square$

## C  Lemma for the Loss at the Last Iteration

We show that if the weights $\boldsymbol{w}^t$ generated in Algorithm 1 converge, then the one-step value $V_{\boldsymbol{w}^T, \boldsymbol{v}^T}$ also converges to the average value of the best weight. This result justifies only considering the last weight given by Algorithm 1.

**Lemma 1.** *If $\forall \epsilon > 0, \exists T_0, \forall t_1, t_2 > T_0, \|\boldsymbol{w}^{t_1} - \boldsymbol{w}^{t_2}\| < \epsilon$, then the loss of the min player's strategy converges to the average loss of an optimal fixed strategy:*

$$\lim_{T \to \infty} V_{\boldsymbol{w}^T, \boldsymbol{v}^T} = \lim_{T \to \infty} \frac{1}{T} \min_{\tilde{\boldsymbol{w}} \in \Delta_n^c} \sum_{t=1}^{t=T} V_{\tilde{\boldsymbol{w}}, \boldsymbol{v}^t} \tag{23}$$

*Proof.*

$$
\begin{aligned}
\frac{1}{T} \sum_{t=1}^{t=T} V_{\boldsymbol{w}^t, \boldsymbol{v}^t} &= \frac{1}{T} \sum_{t=1}^{t=T} V_{\boldsymbol{w}^t, \boldsymbol{v}^t} \\
&= \frac{1}{T} \Big( \sum_{t=1}^{t=T_0} V_{\boldsymbol{w}^t, \boldsymbol{v}^t} + \sum_{t=T_0}^{t=T} V_{\boldsymbol{w}^t, \boldsymbol{v}^t} \Big) \\
&= \frac{1}{T} \Big( \sum_{t=1}^{t=T_0} V_{\boldsymbol{w}^t, \boldsymbol{v}^t} + \sum_{t=T_0}^{t=T} \boldsymbol{w}^t \cdot \boldsymbol{v}^t \Big) \\
&= \frac{1}{T} \Big( \sum_{t=1}^{t=T_0} V_{\boldsymbol{w}^t, \boldsymbol{v}^t} + \sum_{t=T_0}^{t=T} \boldsymbol{w}^T \cdot \boldsymbol{v}^t + \sum_{t=T_0}^{t=T} (\boldsymbol{w}^t - \boldsymbol{w}^T) \cdot \boldsymbol{v}^t \Big)
\end{aligned}
\tag{24}
$$

$$
\begin{aligned}
\lim_{T \to \infty} \frac{1}{T} \sum_{t=1}^{t=T} V_{\boldsymbol{w}^t, \boldsymbol{v}^t} &= \lim_{T \to \infty} \frac{1}{T} \Big( \sum_{t=1}^{t=T_0} V_{\boldsymbol{w}^t, \boldsymbol{v}^t} + \sum_{t=T_0}^{t=T} \boldsymbol{w}^T \cdot \boldsymbol{v}^t + \sum_{t=T_0}^{t=T} (\boldsymbol{w}^t - \boldsymbol{w}^T) \cdot \boldsymbol{v}^t \Big) \\
&= \lim_{T \to \infty} \frac{1}{T} \sum_{t=T_0}^{t=T} \boldsymbol{w}^T \cdot \boldsymbol{v}^t + \lim_{T \to \infty} \frac{1}{T} \Big( \sum_{t=1}^{t=T_0} V_{\boldsymbol{w}^t, \boldsymbol{v}^t} + \sum_{t=T_0}^{t=T} (\boldsymbol{w}^t - \boldsymbol{w}^T) \cdot \boldsymbol{v}^t \Big) \\
&= \lim_{T \to \infty} \boldsymbol{w}^T \cdot \boldsymbol{v}^T + \lim_{T \to \infty} \frac{1}{T} \Big( \sum_{t=1}^{t=T_0} V_{\boldsymbol{w}^t, \boldsymbol{v}^t} + \sum_{t=T_0}^{t=T} (\boldsymbol{w}^t - \boldsymbol{w}^T) \cdot \boldsymbol{v}^t \Big) \\
&= \lim_{T \to \infty} \boldsymbol{w}^T \cdot \boldsymbol{v}^T + 0 = \lim_{T \to \infty} \boldsymbol{w}^T \cdot \boldsymbol{v}^T = \lim_{T \to \infty} V_{\boldsymbol{w}^T, \boldsymbol{v}^T}
\end{aligned}
$$

Therefore, as $T$ approaches $\infty$, the loss at the last epoch $\boldsymbol{w}^T \cdot \boldsymbol{v}^T$ is equal to the average loss $\frac{1}{T} \sum_{t=1}^{t=T} V_{\boldsymbol{w}^t, \boldsymbol{v}^t}$. Then using the upper bound derived in Theorem 1, we can conclude that the final loss $\boldsymbol{w}^T \cdot \boldsymbol{v}^T$ also has an upper bound as $T$ approaches $\infty$.

$$
\begin{aligned}
\lim_{T \to \infty} V_{\boldsymbol{w}^T, \boldsymbol{v}^T} &= \lim_{T \to \infty} \frac{1}{T} \sum_{t=1}^{t=T} V_{\boldsymbol{w}^t, \boldsymbol{v}^t} \\
&\leq \lim_{T \to \infty} \frac{1}{T} \min_{\tilde{\boldsymbol{w}} \in \Delta_n^c} \sum_{t=1}^{t=T} V_{\tilde{\boldsymbol{w}}, \boldsymbol{v}^t} + \frac{\ln n}{T} + (1 + \max_t \alpha_t) \cdot \sqrt{\frac{\ln n}{T}} \\
&= \lim_{T \to \infty} \frac{1}{T} \min_{\tilde{\boldsymbol{w}} \in \Delta_n^c} \sum_{t=1}^{t=T} V_{\tilde{\boldsymbol{w}}, \boldsymbol{v}^t}
\end{aligned}
\tag{25}
$$

Since $\lim_{T \to \infty} \frac{1}{T} \min_{\tilde{\boldsymbol{w}} \in \Delta_n^c} \sum_{t=1}^{t=T} V_{\tilde{\boldsymbol{w}}, \boldsymbol{v}^t}$ is the lower bound for $\lim_{T \to \infty} V_{\boldsymbol{w}^T, \boldsymbol{v}^T}$,

$$\lim_{T \to \infty} V_{\boldsymbol{w}^T, \boldsymbol{v}^T} \geq \lim_{T \to \infty} \frac{1}{T} \min_{\tilde{\boldsymbol{w}} \in \Delta_n^c} \sum_{t=1}^{t=T} V_{\tilde{\boldsymbol{w}}, \boldsymbol{v}^t}. \tag{26}$$

Then from Equation (25) and Equation (26), we obtain

$$\lim_{T \to \infty} V_{\boldsymbol{w}^T, \boldsymbol{v}^T} = \lim_{T \to \infty} \frac{1}{T} \min_{\tilde{\boldsymbol{w}} \in \Delta_n^c} \sum_{t=1}^{t=T} V_{\tilde{\boldsymbol{w}}, \boldsymbol{v}^t}. \tag{27}$$

$\square$

## D    Comparison of our Method and APStar

**Can a fair classification method developed for group fairness be applied to reduce the class-dependent effects of data augmentation?**   Most fair ML methods do not apply to our problem, but some could potentially be adapted (e.g., APStar (Martinez et al., 2020)). As such, we investigate how it performs here compared to our proposition. They achieve fairness by adding a vector with ones for the worst groups to the previous weight vector. Adapting their method involves replacing groups by classes. In our setting, some classes inherently exhibit lower accuracy due to the increased difficulty in classifying them, and APStar tends to over-emphasize these difficult classes while neglecting others, impeding convergence during training. Additionally, as the number of classes grows, there is a greater chance that marginally better classes will be under-weighted and consequently ignored, which further hurts training. As shown in Figure 5, our experiments show that the APStar's weight vector has difficulty converging with many classes. Comparing our method with APStar on the CIFAR-100 dataset, the weight vector and test accuracies of APStar show less stability. We conclude that our method is more effective than APStar as the number of classes grows.

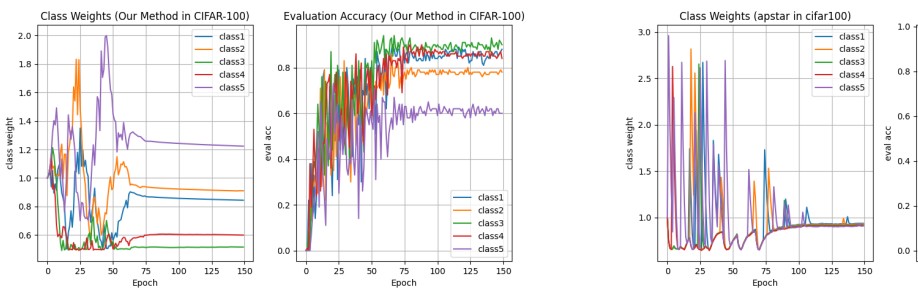

Figure 5: Comparison of class weights and test accuracies between our method and APStar on CIFAR-100 dataset. These results show that the APStar's weight vector has difficulty converging with a large number of classes.

## E    Detailed Descriptions and Hyperparameters of Baselines

**cross entropy loss**   The first baseline is normal cross entropy loss.

$$CE(p_t) = -\ln(p_t) \tag{28}$$

**focal loss**   This method adds a factor to the standard cross entropy loss.

$$FL(p_t) = -\ln(p_t) * (1 - p_t)^\gamma \tag{29}$$

where $p_t$ is the model's estimated probability for the class with label $y = 1$ and $\gamma > 0$ is a hyperparameter.

**tilted cross entropy loss** This method assigns a weight to different classes. The weight of each class is obtained from the previous training loss of this class by the following equation.

$$CE(p_t) = -\ln(p_t) * w_t$$
$$w_t(c) = (1-\gamma)w_{t-1}(c) + \gamma \frac{e^{L_t(c)}}{\sum_{i=1}^{n} e^{L_t(i)}} \tag{30}$$

where $w_t(i)$ is the weight assigned to class $i$ and $L_t(i)$ is the training loss of class $i$. $w_0(i)$ is initialized as $\frac{1}{n}$ for each class $i$.

**performance weighted loss (PW Loss)** Similar to focal loss, this method adds a factor to the standard cross entropy loss.

$$FL(p_t) = -\ln(p_t) * ((1-p_t)^\gamma + \theta) \tag{31}$$

where $\theta$ is an additional hyperparameter that makes this loss more flexible than focal loss .

**GGF enhanced cross entropy loss** This approach utilizes the GGF loss at a frequency specified by $f$. When $f = 1$, the GGF loss is consistently applied. For $f = 2$, the model alternates between applying the standard loss and the GGF loss. The weights assigned to different classes are determined based on the ranking of their training accuracies of the last epoch. The weight for class $i$ at epoch $t$ can be written as:

$$w_t^i = \max[\alpha^{Rank(v_{t-1}^i)-1}, w_{min}] \tag{32}$$

where $v_{t-1}^i$ is the training accuracy of class $i$ at epoch $t-1$, $\alpha$ and $w_{min}$ are two hyperparameters.

Table 5: Hyperparameters of Baselines

| Focal Loss | CIFAR-10 | CIFAR-100 | Fashion-Mnist | Mini-Imagenet | Imagenet |
|---|---|---|---|---|---|
| $\gamma$ | 2.0 | 2.0 | 2.0 | 2.0 | 2.0 |
| TCE Loss | CIFAR-10 | CIFAR-100 | Fashion-Mnist | Mini-Imagenet | Imagenet |
| $\gamma$ | 0.5 | 0.5 | 0.5 | 0.5 | 0.5 |
| PW Loss | CIFAR-10 | CIFAR-100 | Fashion-Mnist | Mini-Imagenet | Imagenet |
| $\gamma$ | 2.5 | 2.5 | 2.5 | 2.5 | 2.5 |
| $\theta$ | 0.8 | 0.8 | 0.8 | 0.8 | 0.8 |
| GGF Loss | CIFAR-10 | CIFAR-100 | Fashion-Mnist | Mini-Imagenet | Imagenet |
| $\alpha$ | 0.9 | 0.98 | 0.98 | 0.95 | 0.998 |
| $w_{\min}$ | 0.1 | 0.1 | 0.1 | 0.01 | 0.2 |
| $f$ | 1 | 2 | 2 | 2 | 1 |

## F Preprocessing of Datasets

For CIFAR-10, CIFAR-100 and Fashion-Mnist datasets, we use the built-in datasets from the library of *torchvision.datasets*. For the Mini-Imagenet and Imagenet datasets, we retrieve them from their respective websites and utilize the predefined partitions for training and evaluation.

## G Hyperparameters of Our Method

There are only two hyperparameters in our method, the parameter of multiplicative weight update $\tau$ and the lower bound of class weights $w_{\min}$. The simplicity in the number of hyperparameters not only eases the implementation of our method but also reduces the complexity of hyperparameter tuning to achieve optimal

performance. Among all five datasets, we do very little tuning and set $\tau = 1$ and $u_{\min} = \frac{1}{2n}$ where $n$ is the number of classes in that specific dataset.

Note that in our theoretical proof appendix B, we set $\tau$ to be $\ln(1 + \frac{1}{1+\max_t \alpha_t}\sqrt{\frac{\ln n}{T}})$. However, it is important to mention that the robustness of our approach is not solely contingent upon this exact value. In practice,we have observed that even when $\tau$ is set to 1, a commonly used default value in experiments, our method still demonstrates strong performance. The consistency of the performance across five datasets underscores our method's resilience and adaptability, indicating that the benefits of our approach are not exclusively tied to the precise tuning of hyperparameters. Results of our method with different $\tau$ values are provided in Appendix L.

## H  Mathematical Definition of Evaluation Metrics

Mathematically, we define the range $r_t$, standard deviation $\sigma_t$ and the coefficient of variation $c_t$ of class accuracies $v_t$ at epoch $t$ as follows.

$$
\begin{aligned}
r_t &= \max_i v_t^i - \min_i v_t^i \\
\mu_t &= \sum_{i=1}^{n} v_t^i \\
\sigma_t &= \sqrt{\frac{\sum_{i=1}^{n}(v_t^i - \mu_t)^2}{n-1}} \\
c_t &= \frac{\sigma_t}{\mu_t}
\end{aligned}
\tag{33}
$$

## I  Plots of Worst Class Accuracies

For an in-depth analysis of the performance improvement in the worst class accuracies, detailed plots are shown in fig. 6, fig. 7, fig. 8, fig. 9 and fig. 10 for five datasets. In each of the five figures, the x-axis represents the $x^{th}$ worst class, while the y-axis denotes the evaluation accuracy of that particular class. This arrangement allows for a clear visualization of the effectiveness of our method in improving the test accuracy of these worst classes within each dataset. By examining the figures, we can observe how our approach consistently outperforms the other baseline methods across different datasets and almost all worst classes, highlighting its robustness and effectiveness in improving the worst class performance.

**Analysis of Worst Class Performance**  Specifically, we investigated whether CLAM identifies the same worst class as other methods and examined the training and testing errors for these classes. We observe that in CIFAR-10 all methods identify "cat" and "dog" as the worst classes while in CIFAR-100 "boy", "girl", "man", "woman", "seal", and "otter" are identified as the worst classes. Notably, while the training accuracy for these classes is always 100%, indicating perfect fitting to the training data, CLAM improves the testing accuracy for these worst classes, thereby reducing the gap between training and testing performance and highlighting CLAM's effectiveness in reducing class-specific generalization gap.

## J  Plots of Fairness Metrics during Training

As shown in fig. 11, fig. 12, fig. 13, fig. 14 and fig. 15, we include plots of fairness metrics during training in CIFAR-10, CIFAR-100, Fashion-Mnist, Mini-ImageNet and ImageNet datasets. From the plots, we can observe that our method consistently outperform other methods with the lowest standard deviation of class accuracies and the lowest coefficient of variation (COV).

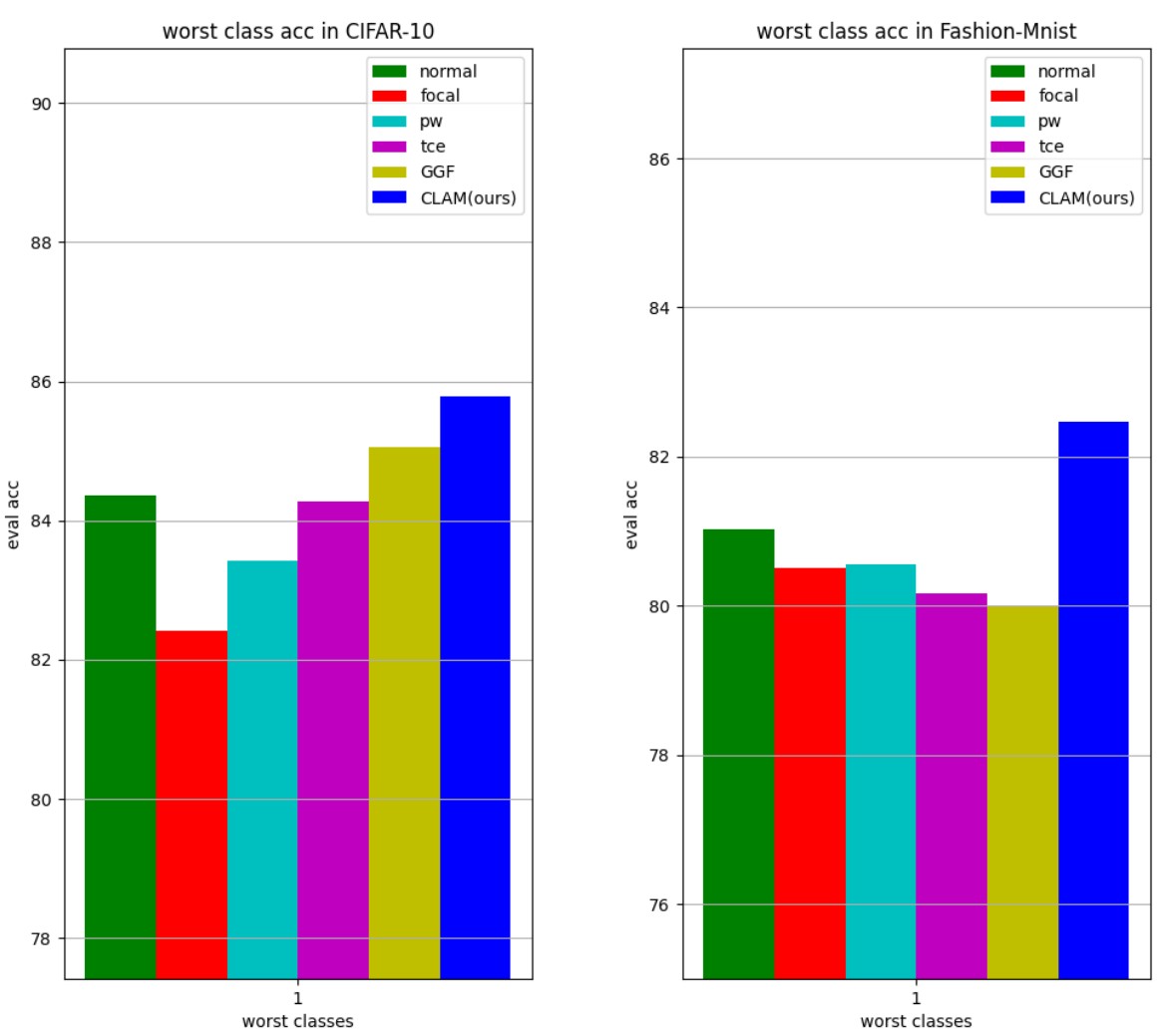

Figure 6: Worst class accuracies of different methods in CIFAR-10 dataset.

Figure 7: Worst class accuracies of different methods in Fashion-Mnist dataset.

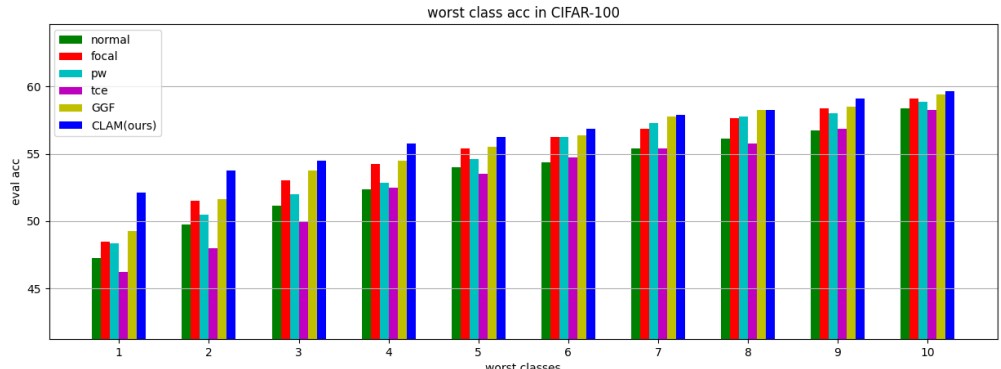

Figure 8: Worst class accuracies of different methods in CIFAR-100 dataset.

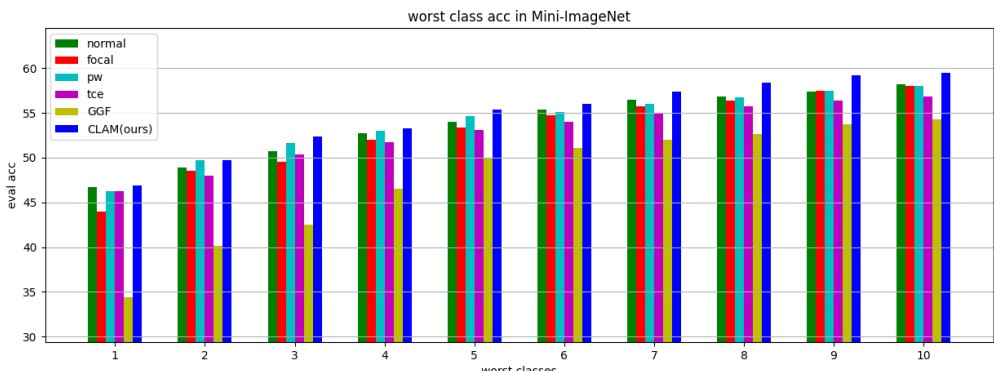

Figure 9: Worst class accuracies of different methods in Mini-Imagenet dataset.

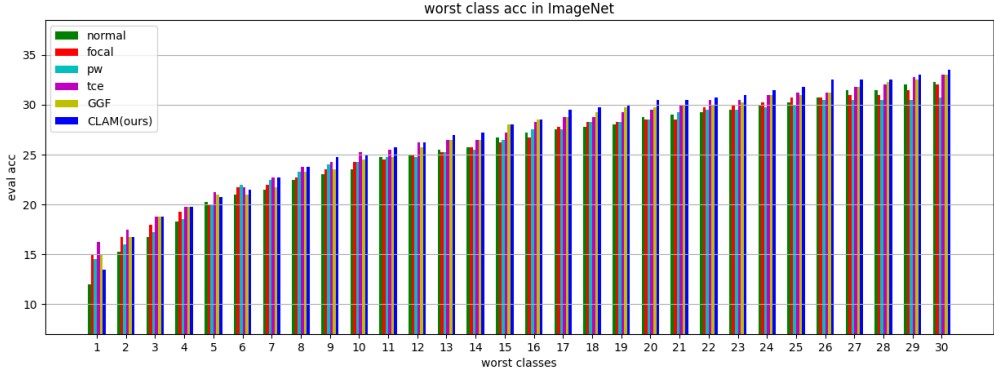

Figure 10: Worst class accuracies of different methods in ImageNet dataset.

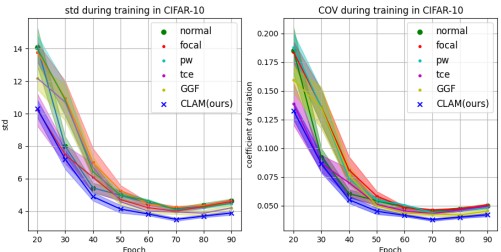

Figure 11: Fairness metrics during training in CIFAR-10 dataset.

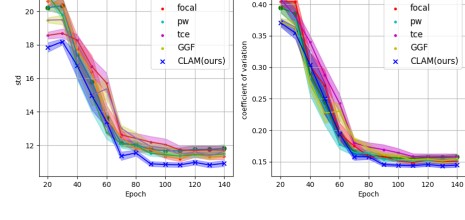

Figure 12: Fairness metrics during training in CIFAR-100 dataset.

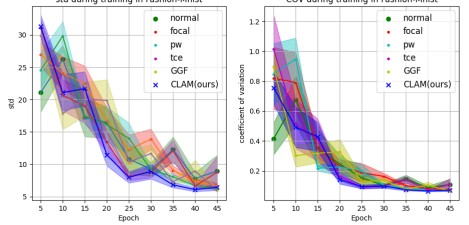

Figure 13: Fairness metrics during training in Fashion-Mnist dataset.

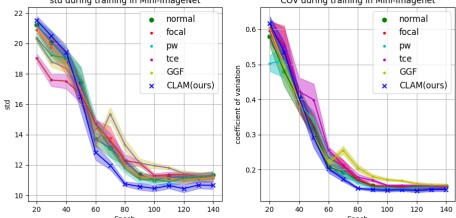

Figure 14: Fairness metrics during training in Mini-ImageNet dataset.

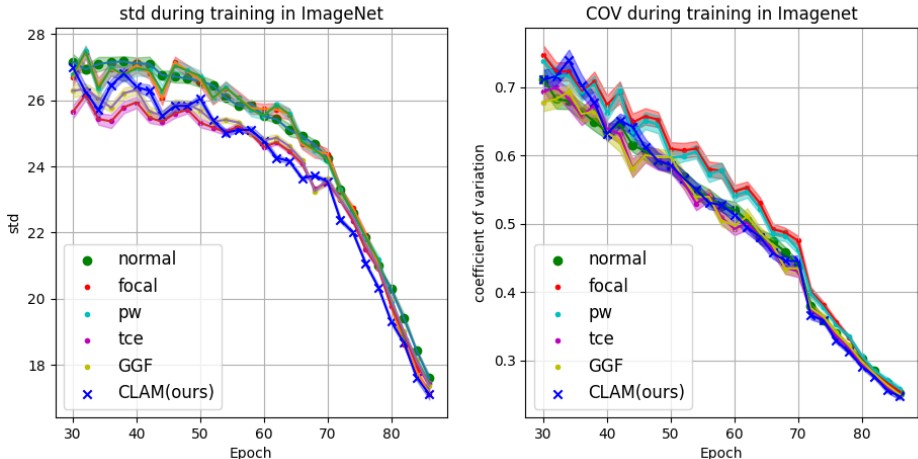

Figure 15: Fairness metrics during training in ImageNet dataset.

# K  Plots of Class Weights Evolution during Training

Here we present the plots depicting the evolution of class weights during training for the CIFAR-100, Fashion-Mnist and Mini-Imagenet datasets in Figure 17, Figure 18 and Figure 19. From these figures, we can discern trends similar to those observed in Figure 16 for CIFAR-10 dataset. Initially, the order of weights fluctuates during training but eventually converges at a certain point, after which the evaluation accuracies also converge. Furthermore, the order of the converged weights is inversely related to the order of the converged evaluation accuracies.

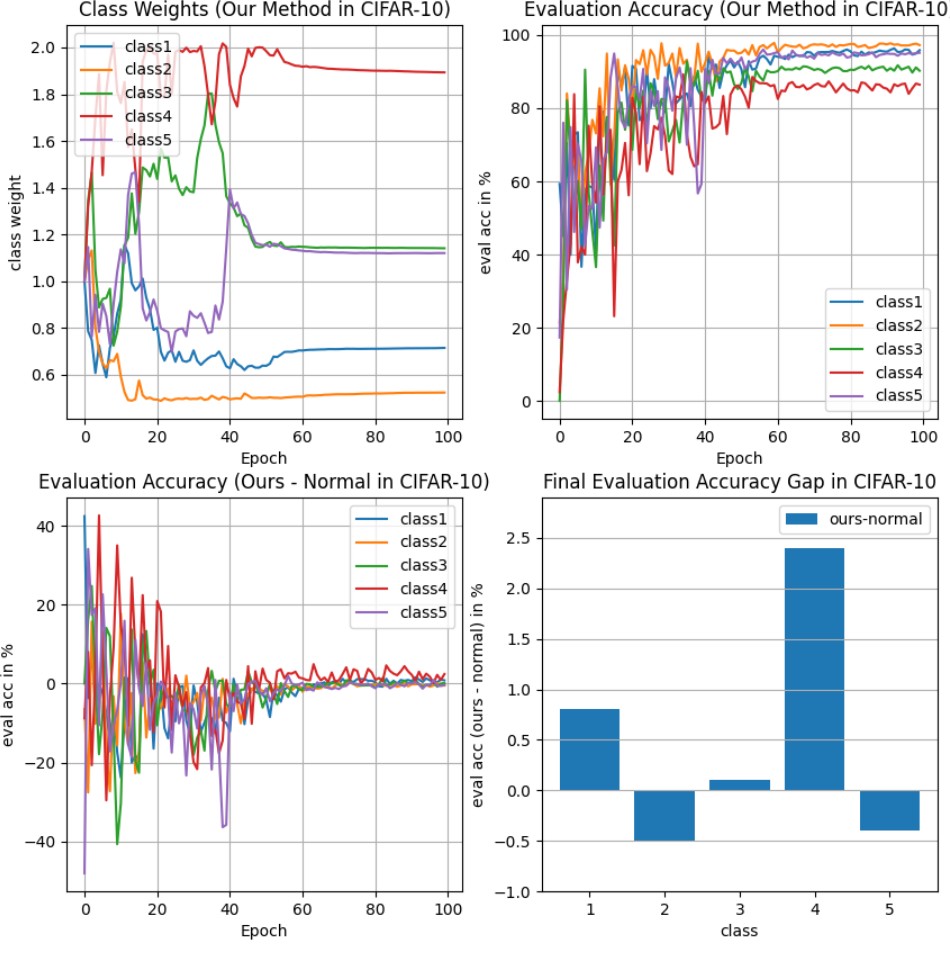

Figure 16: Evolution of class weights in CIFAR-10 dataset. Top Left: class weights; Top Right: evaluation accuracies of certain classes; Bottom Left: evaluation accuracies of certain classes (ours - baseline); Bottom Right: Final evaluation accuracy gap of certain classes (ours - baseline).

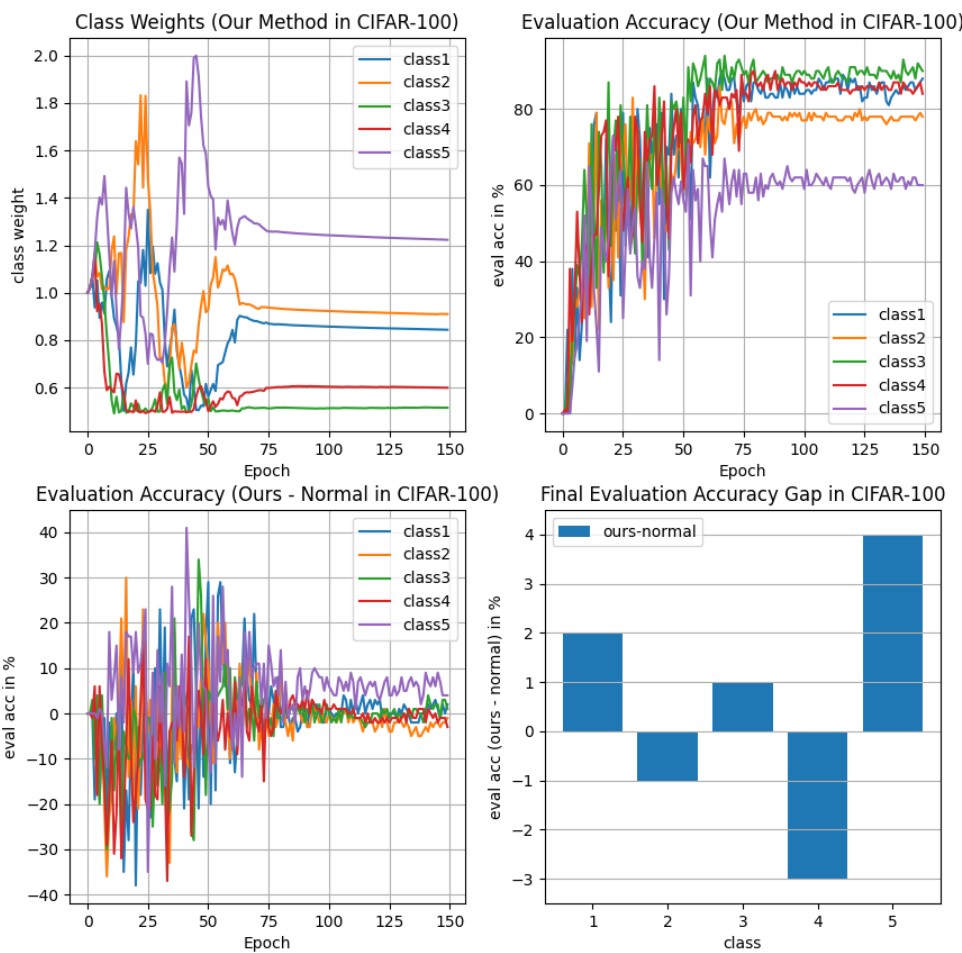

Figure 17: Evolution of class weights in CIFAR-100 dataset. Top Left: class weights; Top Right: evaluation accuracies of certain classes; Bottom Left: evaluation accuracies of certain classes (ours - baseline); Bottom Right: Final evaluation accuracy gap of certain classes (ours - baseline).

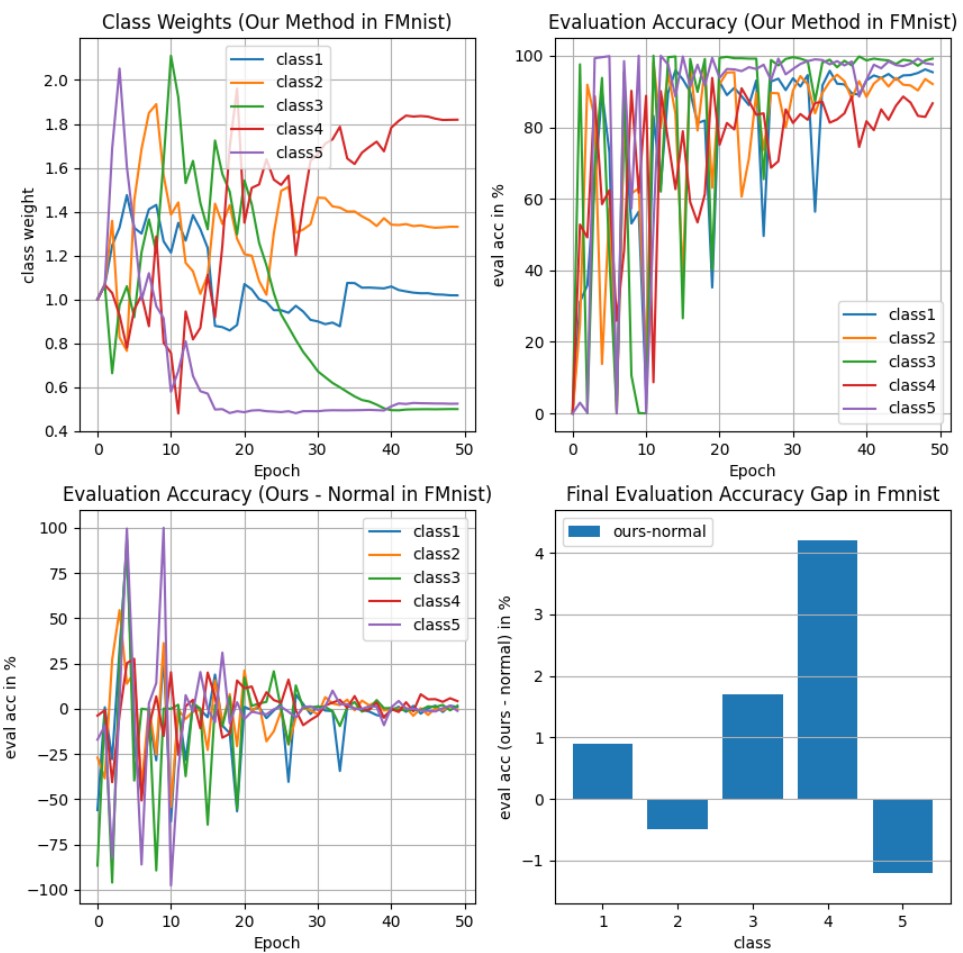

Figure 18: Evolution of class weights in Fashion-Mnist dataset. Top Left: class weights; Top Right: evaluation accuracies of certain classes; Bottom Left: evaluation accuracies of certain classes (ours - baseline); Bottom Right: Final evaluation accuracy gap of certain classes (ours - baseline).

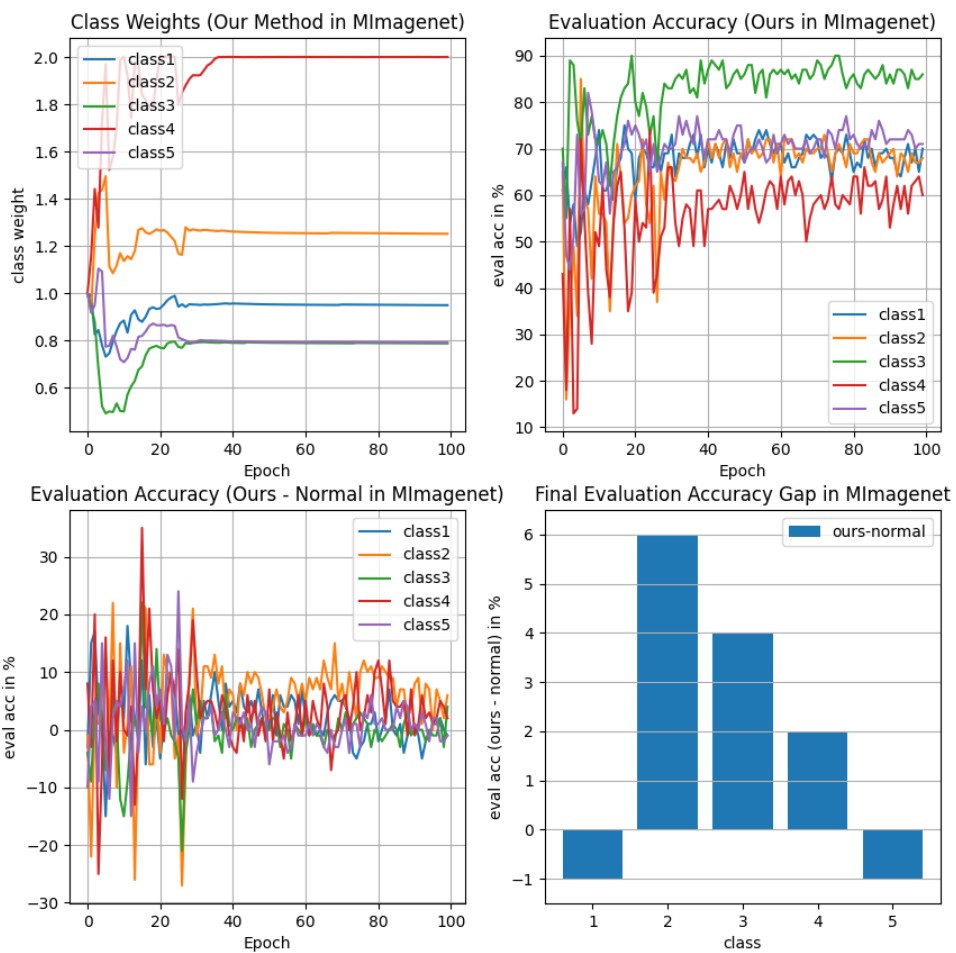

Figure 19: Evolution of class weights in Mini-Imagenet dataset. Top Left: class weights; Top Right: evaluation accuracies of certain classes; Bottom Left: evaluation accuracies of certain classes (ours - baseline); Bottom Right: Final evaluation accuracy gap of certain classes (ours - baseline).

## L  Additional Results for CLAM with Different Values of $\tau$

We present additional results for CLAM on CIFAR-100 and Fashion-Mnist with varying $\tau$ values in Table 6, including the baseline ($\tau = 0$) for reference. Across both datasets, CLAM consistently improves class equity in terms of "std", "COV", "range", and "worst acc". Notably, optimal performance is achieved at $\tau = 0.1$ or $\tau = 0.5$, aligning with theoretical values derived in Theorem 1. For CIFAR-100 (100 classes, 150 training epochs), the theoretically optimal $\tau \approx 0.15$, and for Fashion-Mnist (10 classes, 50 training epochs), the theoretically optimal $\tau \approx 0.18$. The approximations come from assuming $\max_t \alpha_t \approx 0.1$ in both cases.

Table 6: We present the fairness metrics of CLAM with different values of $\tau$ on CIFAR-100 dataset in the following table. In the columns, "std", "COV", "range", 'avg acc', and 'worst acc' respectively denote the standard deviation, coefficient of variation, range of class accuracies, average accuracy of all classes, and average accuracy of worst 10% classes on test datasets. For each method used on each dataset, the reported result is averaged over 8 crop lower bounds. The std, range, and accuracies are listed in % while COV is listed in values. **Lower is better for "std", "range", and "COV". Higher is better for accuracies.**

| $\tau$ | CIFAR-100 | | | | | Fashion-Mnist | | | | |
|---|---|---|---|---|---|---|---|---|---|---|
| | std | COV | range | avg acc | worst acc | std | COV | range | avg acc | worst acc |
| 0 (Normal) | 11.8 | 0.157 | 39.1 | 75.2 | 53.6 | 5.94 | 0.063 | 18.3 | **94.1** | 81.0 |
| 0.1 | 10.8 | 0.142 | 35.9 | **75.8** | 56.0 | **5.17** | **0.055** | **14.8** | **94.1** | **84.7** |
| 0.5 | **10.6** | **0.140** | **35.3** | 75.5 | 56.1 | **5.17** | **0.055** | 15.1 | **94.1** | 84.4 |
| 1.0 | 10.8 | 0.143 | 35.9 | 75.7 | **56.4** | 5.44 | 0.058 | 16.7 | 94.0 | 82.5 |
| 2.0 | 11.3 | 0.151 | 37.3 | 75.1 | 54.5 | 5.30 | 0.057 | 15.4 | 93.9 | 83.7 |
| 10.0 | 11.3 | 0.151 | 37.3 | 75.1 | 54.9 | 5.50 | 0.060 | 17.1 | 92.8 | 81.7 |

Table 7: We present the fairness metrics of different methods with color jitter as data augmentation on four datasets in the following table. In the second column of the table, "std", "COV", and "range" respectively denote the standard deviation, coefficient of variation, and range of class accuracies on test datasets. For each method used on each dataset, the reported result is averaged over 8 color jitter parameters. The std and range are listed in % while COV is listed in values. **Lower is better.**

| Method | CIFAR-10 | | | CIFAR-100 | | | Fashion-Mnist | | | Mini-ImageNet | | |
|---|---|---|---|---|---|---|---|---|---|---|---|---|
| | std | COV | range | std | COV | range | std | COV | range | std | COV | range |
| Normal | 8.95 | 0.106 | 28.7 | 16.89 | 0.273 | 57.1 | 15.8 | 0.238 | 48.7 | 17.4 | 0.312 | 58.2 |
| Focal | 8.66 | 0.102 | 26.4 | 15.27 | 0.242 | 51.3 | 15.7 | 0.218 | 46.9 | 17.3 | 0.326 | 58.5 |
| GGF | 8.88 | 0.104 | 25.9 | 17.72 | 0.308 | 59.0 | 15.9 | 0.225 | 47.4 | 22.0 | 0.492 | 73.5 |
| **CLAM (Ours)** | **3.86** | **0.042** | **24.3** | **15.06** | **0.236** | **50.1** | **15.5** | **0.214** | **44.7** | **16.3** | **0.290** | **54.1** |

## M  Additional Results with Color Jitter as Data Augmentation

We present additional results on CIFAR-10, CIFAR-100, Fashion-MNIST, and Mini-Imagenet datasets with color jitter as a data augmentation method in Table 7. Our method consistently achieves the best class equity across all datasets, demonstrating its robustness to diverse augmentation strategies.

## N  Additional Results on iNaturalist2018

Here we include additional results on iNaturalist2018, a class-imbalanced dataset, as shown in Table 8. As shown in the table, "Focal" and "GGF" reduce accuracy standard deviation and range but sacrifice significant average accuracy. In contrast, our method achieves a lower coefficient of variation, a more advanced and critical metric for class equity, while maintaining better average, worst-class, and best-class accuracies, demonstrating stronger robustness and balance across classes.

Table 8: We present the fairness metrics of different methods with random resized cropping as data augmentation on iNaturalist2018 in the following table. In the second column of the table, "std", "COV", "range", "avg acc", "worst acc", and "best acc" respectively denote the standard deviation, coefficient of variation, range of class accuracies, average accuracy of all classes, average accuracy of worst 10% classes, and average accuracy of best 10% classes on test datasets. For each method used on each dataset, the reported result is averaged over 8 crop lower bounds. The std, range, and accuracies are listed in % while COV is listed in values. **Lower is better for "std", "range", and "COV". Higher is better for accuracies.**

| iNaturalist2018 | | | | | | |
|---|---|---|---|---|---|---|
| Method | std | COV | range | avg acc | worst acc | best acc |
| Normal | 30.5 | 0.843 | 50.7 | 42.1 | 11.1 | 61.9 |
| Focal | 29.3 | 0.933 | **47.9** | 37.9 | 8.3 | 56.2 |
| GGF | **29.2** | 0.875 | 48.1 | 40.2 | 9.3 | 57.5 |
| **CLAM (Ours)** | 30.3 | **0.813** | 50.2 | **42.9** | **12.5** | **62.7** |

Table 9: We present the fairness metrics of different methods on ImageNet with reassessed labels in the following table. In the second column of the table, "std", "COV", "range", "worst acc", and "best acc" respectively denote the standard deviation, coefficient of variation, range of class accuracies, and average accuracy of worst 10% classes on test datasets. For each method used on each dataset, the reported result is averaged over 8 crop lower bounds. The std, range, and accuracies are listed in % while COV is listed in values. **Lower is better for "std", "range", and "COV". Higher is better for accuracies.**

| ImageNet with ReaL labels | | | |
|---|---|---|---|
| Method | std | COV | range | worst acc |
| Normal | 13.6 | 0.179 | 45.9 | 50.0 |
| **CLAM (Ours)** | **13.3** | **0.176** | **44.9** | **50.2** |

## O  Additional Results on Imagenet with ReaL (Beyer et al., 2020; Kirichenko et al., 2023)

Here we include additional results on Imagenet dataset with reassessed labels, as shown in Table 9. From the table, we can observe that our method continues to improve class equity on ImageNet with reassessed labels.

## P  Compute Resources We Use

In all our experiments, we utilize a GPU server equipped with 8 cards that have either RTX-4090 or A6000 GPUs and are powered by AMD EPYC 7763 CPUs. For the CIFAR-10 dataset experiments, training a resnet-50 model for 100 epochs on a single GPU card takes approximately 3 hours. When working with the CIFAR-100 dataset, a resnet-50 model trained for 150 epochs on one GPU card requires about 4.5 hours. On the Fashion-MNIST dataset, a resnet-50 model trained for 50 epochs on one GPU card completes in less than 2 hours. For the Mini-Imagenet dataset, an experiment involving a resnet-50 model trained for 150 epochs on one GPU card consumes around 4.5 hours. Lastly, for the Imagenet dataset, training a resnet-50 model for 88 epochs on one GPU card demands roughly 2 days of computational time.

# Q   Limitations

While our approach has yielded commendable improvements in fairness metrics, it is essential to acknowledge a limitation that emerged from our experiments. We observe a small decrease in overall test accuracy across 3 out of the 5 datasets we examined, as discussed in Table 4. This trade-off between fairness and accuracy presents a limitation of our work, as enhancing fairness may sometimes come at the expense of the overall model performance.

# R   Broader Impacts

**Positive societal impacts**   By proposing and validating a novel method that balances accuracy with fairness, we contribute to the development of a more equitable machine learning system. This kind of system have the potential to positively affect various domains, such as healthcare and education, where unbiased decisions can lead to more just outcomes for all individuals.

**Negative societal impacts**   To the best of our knowledge, we don't see any negative societal impacts of our work.

