# OpenReview forum: "Understanding and Reducing the Class-Dependent Effects of Data Augmentation with A Two-Player Game Approach"
_TMLR — Accepted by TMLR_

### Review · Reviewer_mh54 · 2025-04-15

**Summary Of Contributions:**

This paper addresses the issue of inherent class fairness introduced by data augmentation. The authors propose CLAM (CLAss-dependent Multiplicative-weights), a novel framework as an adversarial two-player game, where the max player (Eq 3) aims to maximize the overall performance across different classes, and the min player (Eq 4) aims to reweight more on the worst-case class. The authors theoretically prove the convergence of CLAM under constrained simplex settings and empirically validate their method on five datasets.

**Audience:**

Yes

**Claims And Evidence:**

Yes

**Requested Changes:**

Please refer to my comments in the section of disadvantages.

**Strengths And Weaknesses:**

Strengths:

1. CLAM is easy to implement, requires few hyperparameters, and is architecture-agnostic.

2. The authors provide a theoretical guarantee for CLAM.

Weaknesses:

1. My primary concern is that this paper mentioned the effect of data augmentation on class fairness, but the proposed method is reweighting each sample in a two-player game way. Could the authors explain more about how it is related to data augmentation? and explain why it can solve the bias between the accuracies among different classes? For example, as mentioned in the first paragraph of the Introduction, data augmentation for image classification includes random shifts and random clips. Why this semantic-preserving transformation can induce class bias? Why CLAM can alleviate it?


2. Based on the first disadvantage, if I understand correctly, I do not think the authors involve the effect of data augmentation in the theoretical analysis of CLAM. Usually we consider data augmentation as a label-invariant function. Please refer to [1].


3. The datasets utilized in the experiments are all class-balanced datasets, but class bias occurs more significantly via class-imbalanced datasets (e.g., iNaturalist[2]). I suggest authors add more experiments on imbalanced datasets.

4. The baselines shown in Tables 2/3/4 are quite old (2017-2022). Can the authors compare CLAM with more recent baselines for class fairness?

5. In Table 3, the performance gain for the worst-case accuracy is quite limited on CIFAR10. I suggest the authors conduct more analysis on that. For example, does CLAM have the same worst class as other methods? What is the worst class actually? As the basic idea for CLAM is to assign more weight to the worst-case class, what is the training error for the worst class? By comparing the training error and testing error for the worst class, the authors can identify whether the error is brought by insufficient optimization or the generalization gap.

[1] Yang S, Dong Y, Ward R, et al. Sample efficiency of data augmentation consistency regularization[C]//International Conference on Artificial Intelligence and Statistics. PMLR, 2023: 3825-3853.

[2] https://paperswithcode.com/dataset/inaturalist

---

> ### Author Response · Authors · 2025-05-12
> **Rebuttal to Reviewer mh54**
>
> We thank the reviewer for saying that our proposed method, CLAM, is easy to implement, requires few hyperparameters, and is architecture-agnostic. Below we address the reviewer's concerns one by one.
>
> 1. "My primary concern is that this paper mentioned the effect of data augmentation on class fairness, but the proposed method is reweighting each sample in a two-player game way. Could the authors explain more about how it is related to data augmentation? and explain why it can solve the bias between the accuracies among different classes? For example, as mentioned in the first paragraph of the Introduction, data augmentation for image classification includes random shifts and random clips. Why this semantic-preserving transformation can induce class bias? Why CLAM can alleviate it?"
> 2. "Based on the first disadvantage, if I understand correctly, I do not think the authors involve the effect of data augmentation in the theoretical analysis of CLAM. Usually we consider data augmentation as a label-invariant function."
>
> Here we answer the first and second questions together. Our work is motivated by class-dependent effects of data augmentation. To address, we formalize the classification problem as a fair optimization problem, as detailed in Section 4. However, by contrasting the traditional formulation (Equation (3)) with our proposed one (Equation (5)), we demonstrate that class equity is not solely a consequence of data augmentation but a more general problem. Recognizing the broader nature of this problem, we adopt a more global solution framework that dynamically balances class weights to mitigate disparities, ensuring equitable accuracy without compromising overall performance.
>
> 3. "The datasets utilized in the experiments are all class-balanced datasets, but class bias occurs more significantly via class-imbalanced datasets (e.g., iNaturalist). I suggest authors add more experiments on imbalanced datasets."
>
> We thank the reviewers for suggesting additional experiments to address the class equity in imbalanced datasets (e.g., iNaturalist2018). In response, we have included results of "Normal" and our proposed method, "CLAM", for iNaturalist2018 in the revised version (see Table 7 in Appendix L). Experiments for other baseline methods on iNaturalist2018 are currently underway and will be included in the final submission. Current experimental results demonstrate that "CLAM" effectively achieves equitable performance even in highly imbalanced settings, consistent with our findings on balanced datasets. This further validates the robustness and generality of our approach across diverse data distributions.
>
> 4. "The baselines shown in Tables 2/3/4 are quite old (2017-2022). Can the authors compare CLAM with more recent baselines for class fairness?"
>
> We appreciate the reviewer’s feedback regarding the baselines and acknowledge the value of incorporating more recent methods for class fairness. In response, we are expanding our experiments to include relabeling-based baselines [1], which offers an alternative approach to reducing class-dependent effects of data augmentation. For comparison, we have also integrated CLAM within the relabeling framework. These updated results will be included in the final version.
>
>     [1] Polina Kirichenko, Mark Ibrahim, Randall Balestriero, Diane Bouchacourt, Ramakrishna Vedantam, Hamed Firooz, and Andrew Gordon Wilson. Understanding the detrimental class-level effects of data augmentation, 2023. URL https://arxiv.org/abs/2401.01764.
>
> 5. "In Table 3, the performance gain for the worst-case accuracy is quite limited on CIFAR10. I suggest the authors conduct more analysis on that. For example, does CLAM have the same worst class as other methods? What is the worst class actually? As the basic idea for CLAM is to assign more weight to the worst-case class, what is the training error for the worst class? By comparing the training error and testing error for the worst class, the authors can identify whether the error is brought by insufficient optimization or the generalization gap."
>
> We thank the reviewer for their insightful comments on the limited worst-case accuracy gain observed in Table 3 for CIFAR-10. To address this, we have conducted a deeper analysis of the worst-class performance. Specifically, we investigated whether CLAM identifies the same worst class as other methods and examined the training and testing errors for these classes. We observe that all methods identify "cat" and "dog" as the worst classes in CIFAR-10. Notably, while the training accuracy for these classes is always 100%, indicating perfect fitting to the training data, CLAM improves the testing accuracy for "cat" and "dog", thereby reducing the gap between training and testing performance and highlighting CLAM’s effectiveness in reducing class-specific generalization gap. Detailed worst-class analysis for other datasets will be included in final version to further demonstrate our method's effectiveness.

---

### Review · Reviewer_8JWK · 2025-04-16

**Summary Of Contributions:**

This work proposes a min-max optimization approach to tackle the problem of class-dependent accuracy of the Neural Network classifiers trained with data augmentation methods. Theoretical analysis is provided to justify the validity of the model and empirical evidences are provided to justify the effectiveness of the model.

**Audience:**

Yes

**Claims And Evidence:**

Yes

**Requested Changes:**

Please see weakness.

**Strengths And Weaknesses:**

Pros:

1. Paper is well written and easy to follow.
2. Detailed theoretical analysis are provided.
3. Experimental results look good, justifies the effectiveness of the model

Cons:

1. My major concern is the motivation of the problem. In short, it is not straightforward to me that the problem of Class-Dependent Effects is essential in practical deployment of Neural Network models. And from my reading experience of this paper, the importance of the Class-Dependent Effects is not emphasized (only the autonomous driving example in the intro is not sufficient ). I would recommend the author(s) provide stronger examples that the Class-Dependent Effects are crucial.

2. The effectiveness of the proposed method may be limited. This can be seen In Fig.2, where the per-class accuracy of the proposed method and the corresponding weight of loss seems converge at around 60 epoch. And the final per-class accuracy still some what differs, implying that the proposed method is mitigating (instead of solving) the problem of Class-Dependent Effects . I would recommend to plot the same metric without using the proposed methods in contrast to demonstrate the effectiveness of the model. And it would be better if Fig.2 for larger dataset can be provided, as CIFAR10 is a really small dataset.

3. In the experiment section, it seems that only a particular data augmentation technique -- cropping, is tested. What about other data augmentation methods? As most of the previous sections argues that the proposed method is for Class-Dependent Effects when using data augmentation methods, I would expect at least two common types of data augmentation methods are tested.

---

> ### Author Response · Authors · 2025-05-12
> **Rebuttal to Reviewer 8JWK**
>
> We thank the reviewer for saying that our paper is well written and easy to follow. Below we address the reviewer's concerns one by one.
>
> 1. "My major concern is the motivation of the problem. In short, it is not straightforward to me that the problem of Class-Dependent Effects is essential in practical deployment of Neural Network models. And from my reading experience of this paper, the importance of the Class-Dependent Effects is not emphasized (only the autonomous driving example in the intro is not sufficient). I would recommend the author(s) provide stronger examples that the Class-Dependent Effects are crucial."
>
> We appreciate the reviewer’s insight and acknowledge the need to further emphasize the practical relevance of class-dependent effects. While we initially highlighted autonomous driving as an illustrative example, such effects are equally critical in domains like healthcare, finance, and e-commerce, where misclassifying certain classes (e.g., malignant tumors, fraudulent transactions, or critical product defects) can lead to severe harm despite strong overall model performance. This underscores why addressing class-dependent effects is vital for real-world deployment.
>
> 2. "The effectiveness of the proposed method may be limited. This can be seen In Fig.2, where the per-class accuracy of the proposed method and the corresponding weight of loss seems converge at around 60 epoch. And the final per-class accuracy still some what differs, implying that the proposed method is mitigating (instead of solving) the problem of Class-Dependent Effects . I would recommend to plot the same metric without using the proposed methods in contrast to demonstrate the effectiveness of the model. And it would be better if Fig.2 for larger dataset can be provided, as CIFAR10 is a really small dataset."
>
> We acknowledge the reviewer’s concerns and clarify that while per-class metrics converge, our method significantly narrows class disparities (e.g., lower standard deviation, COV and range in accuracy on CIFAR-10 vs. baseline). To demonstrate effectiveness, we now include baseline comparisons for all datasets in the revised manuscript (see Figs 16, 17, 18, and 19 in Appendix J), showing systematic reductions in class-dependent effects. Larger datasets (CIFAR-100, mini-ImageNet) validate scalability, with the method consistently improving minority class performance, as detailed in the appendix.
>
> 3. "In the experiment section, it seems that only a particular data augmentation technique -- cropping, is tested. What about other data augmentation methods? As most of the previous sections argues that the proposed method is for Class-Dependent Effects when using data augmentation methods, I would expect at least two common types of data augmentation methods are tested."
>
> To address this concern, we have evaluated color jitter as an additional augmentation technique. In the revised version, we now include results for both "Normal" and "CLAM" with color jitter. We are also conducting further experiments to assess other baselines with color jitter, and these results will be incorporated into the final manuscript. Current results across CIFAR-10, CIFAR-100, Fashion-MNIST, and mini-ImageNet (see Table 6 in Appendix K) consistently show that our method reduces class-dependent effects, demonstrating its robustness to diverse augmentation strategies.

---

### Review · Reviewer_Fbdn · 2025-04-29

**Summary Of Contributions:**

This paper presents CLAM (CLAss-dependent Multiplicative-weights), a novel approach to mitigating the class-dependent effects of data augmentation in multi-class classification. It formulates the classification problem as a fair optimization challenge through an adversarial two-player game. The min player assigns weights to classes emphasizing fairness, while the max player optimizes the weighted accuracy. The authors introduce an adapted multiplicative weight algorithm to dynamically balance overall accuracy and class fairness. Theoretical analysis demonstrates the convergence of the proposed algorithm, and extensive empirical evaluations across five datasets confirm its effectiveness in improving worst-class performance with minimal impact on overall accuracy.

**Audience:**

Yes

**Broader Impact Concerns:**

The broader impact statement provided is clear and adequate, highlighting positive societal impacts through fairer decision-making systems and not identifying negative impacts. However, to strengthen it further, the authors could explicitly discuss scenarios where class-level fairness might inadvertently lead to suboptimal outcomes in critical applications, advising caution when deploying fairness-aware methods without domain-specific considerations.

**Claims And Evidence:**

Yes

**Requested Changes:**

- Further Analysis of Accuracy-Fairness Trade-off: Provide deeper insights or a theoretical exploration explaining under which conditions the trade-off between fairness and overall accuracy occurs. This would strengthen the theoretical foundation and provide practical guidelines for applying CLAM effectively.
- Hyperparameter Sensitivity Study: Conduct a detailed analysis of hyperparameter sensitivity. While simplicity is beneficial, understanding the robustness of the approach to hyperparameter choices is crucial for practical adoption.

**Strengths And Weaknesses:**

Strengths
- The paper addresses an important and understudied issue of class-dependent bias introduced by data augmentation. The approach offers a practical solution to ensure equitable performance across classes.
- A robust theoretical justification, including proof of convergence for the multiplicative weight algorithm, is provided.
- Extensive experimentation across multiple datasets and comparison with relevant baselines demonstrate consistent improvements in fairness metrics.
- The proposed method is easy to implement, with only two hyperparameters, making it practical for real-world applications.
- The paper is well-written, clear, and logically organized, facilitating understanding.
Weaknesses
- A slight degradation in overall accuracy observed in some datasets needs more explicit analysis and potential remedies.
- While simplicity is a strength, the impact of hyperparameter tuning is not extensively analyzed, potentially limiting generalizability.
- The iterative updating of weights could introduce computational overhead, which is not thoroughly discussed.

---

> ### Author Response · Authors · 2025-05-12
> **Rebuttal to Reviewer Fbdn**
>
> We thank the reviewer for saying that our paper is well-written, clear, and logically organized.
> Below we address the reviewer's concerns one by one.
>
> 1. "Further Analysis of Accuracy-Fairness Trade-off: Provide deeper insights or a theoretical exploration explaining under which conditions the trade-off between fairness and overall accuracy occurs. This would strengthen the theoretical foundation and provide practical guidelines for applying CLAM effectively."
>
> We appreciate the reviewer's suggestion to delve deeper into the conditions under which the trade-off between fairness and accuracy exists. To the best of our knowledge, the trade-off between these two objectives is inherent and arises from the fact that optimizing for fairness often involves adjusting the model's predictions to ensure equitable treatment across different classes. This adjustment can lead to a slight decrease in overall accuracy because the model may no longer be optimizing purely for the most accurate predictions but rather balancing accuracy with fairness constraints. In our method, assigning more weight to worse-performing classes can lead to a more balanced accuracy distribution but may result in a slight decrease in accuracy for some other classes.
>
> 2. "Hyperparameter Sensitivity Study: Conduct a detailed analysis of hyperparameter sensitivity. While simplicity is beneficial, understanding the robustness of the approach to hyperparameter choices is crucial for practical adoption."
>
> We fully agree with the reviewer that analyzing the sensitivity of our method to hyperparameter choices is critical for its practical adoption. Currently, our method has only one primary hyperparameter, $\tau$, that controls the update rate of the weight vector. To address this, we are now conducting an in-depth hyperparameter sensitivity study, which includes additional experiments with different $\tau$ and will include the results (plots of accuracy and fairness metrics as function of $\tau$) in the final version.

---

### Decision · Action_Editor_tDQ6 · 2025-05-31

**Recommendation:** Accept with minor revision

**Additional Comments:**

As the reviewers mentioned, the motivation of starting from data augmentation is not so clear. From my perspective this is a general methodology for algorithmic fairness and may not be restricted to the fairness for data augmentation. I would like to see some extension or discussion on general algorithmic fairness.

Also, there are some complaints from the reviewers on the experiments. The authors promise to provide additional experiment results with more in-depth analysis. Hence a minor revision is requested.

**Audience:**

Yes

**Audience Explanation:**

I do think this manuscript introduces a novel perspective on algorithmic fairness from the game theory and connect the fairness metric with the concept of regret in online learning that can be achieved with standard online learning algorithms. People work on algorithmic fairness as well as online learning may benefit from this manuscript.

**Claims And Evidence:**

Yes

**Claims Explanation:**

Overall the manuscript is well-written and easy to follow. The proposed method is intuitive with both theoretical and empirical justification. The authors also thoroughly discuss the connection between the proposed methods and other related works.

---

> ### Author Response · Authors · 2025-06-19
>
> Dear AE,
>
> We thank the action editor for the positive feedback on the clarity and readability of our paper.
> Based on the reviewers' suggestions, we have made the following modifications in the camera ready revision:
>
> * In the related work section, we clarify that our method aligns with general algorithmic fairness approaches, as it reduces class-dependent effects of data augmentation while remaining generic and applicable regardless of data augmentation.
> * In response to reviewer feedback, we have included additional experimental results in the appendix.
>     Appendix L presents a hyperparameter sensitivity study for CLAM with varying $\tau$ values.
>     Appendix M includes results of using color jittering as data augmentation.
>     Appendix N provides results on the class-imbalanced iNaturalist2018 dataset.
>     Appendix O shows the comparison of our method with recent baseline.
> * In Appendix C, we add a new lemma justifying only considering the weight at the last iteration instead of the average (over time) of the weights in our case.
>
> Thanks, Authors